# In-Context Principle Learning from Mistakes

## Abstract

In-context learning (ICL, also known as *few-shot prompting*) has been the standard method of adapting LLMs to downstream tasks, by learning from a few input-output examples. Nonetheless, all ICL-based approaches only learn from *correct* input-output pairs. In this paper, we revisit this paradigm, by learning *more* from the few given input-output examples. We introduce Learning Principles (LEAP): First, we intentionally induce the model to *make mistakes* on these few examples; then the model itself reflects on these mistakes, and learn explicit task-specific "principles" from them without any human supervision, which help solve similar problems and avoid common mistakes; finally, we prompt the model to answer unseen test questions using the original few-shot examples and these learned general principles. We evaluate LEAP on a wide range of benchmarks, including multi-hop question answering (Hotpot QA), textual QA (DROP), Big-Bench Hard reasoning, and math problems (GSM8K and MATH); in all these benchmarks, LEAP improves the strongest available LLMs such as `GPT-3.5-turbo`, `GPT-4`, `GPT-4-turbo` and `Claude-2.1`. For example, LEAP improves over the standard few-shot prompting using GPT-4 by 7.5% in DROP, and by 3.3% in HotpotQA. Importantly, LEAP does not require any more input or examples than the standard few-shot prompting settings.

## 1. Introduction

The rise of large language models (LLMs; Radford et al., 2019; Chowdhery et al., 2022; Zhang et al., 2022; Li et al., 2022; Anil et al., 2023; Touvron et al., 2023a;b) that are too costly to finetune for downstream tasks has led to the growing popularity of *in-context learning* (ICL), also known

| Learned Principle | Benchmark |
|---|---|
| When solving a problem involving ==multiple sources of income and expenses, it is crucial to keep track of each component separately== and calculate the total accurately. | GSM8K |
| When simplifying complex numbers raised to powers, it is important to remember the following rules: (1) $i^2 = -1$ (2) $i^3 = -i$ (3) $i^4 = 1$ (4) $i^{-1} = \frac{1}{i} = -i$ | MATH |
| Perform calculations using the ==full precision available and only round as a final step==, if necessary. | DROP |
| When answering questions about ==commonalities== between two entities, it is important to consider ==all relevant aspects== and not just the most obvious or prominent one. | HotpotQA |
| ==Double negation, as in 'not not'==, cancels out and returns the original value. | Boolean Expressions (BBH) |
| ==Sarcasm often involves saying the opposite of what is meant== ... Paying attention to the incongruity between the literal meaning of the words and the intended meaning can help in accurately identifying sarcasm. | Snarks (BBH) |

Figure 1: Examples for learned principles using LEAP, with key idea of each principle ==highlighted.==

as few-shot prompting (Brown et al., 2020; Liu et al., 2023; Wei et al., 2023). In in-context learning, the LLM is provided with a few (e.g., three) input-output task-specific examples in its prompt, along with an unseen test input. Using this emergent ability (Wei et al., 2022b), the LLM is then expected to generate output for the test input. The LLM generates this output by implicitly learning the task from the few given examples, at inference time.

ICL was shown to be extremely effective and data-efficient across a variety of tasks and domains (Min et al., 2022a; Alayrac et al., 2022; Liu et al., 2021; Lu et al., 2023), mainly because it allows for downstream task-adaptation without training. Further, ICL enables generalization using only a few annotated examples. Nonetheless, all ICL-based approaches only learn from *correct* input-output pairs (Min et al., 2022b); in contrast, learning from *mistakes* is a foundational aspect of the way humans learn (Edmondson, 1996;

[1]Anonymous Institution, Anonymous City, Anonymous Region, Anonymous Country. Correspondence to: Anonymous Author <anon.email@domain.com>.

Preliminary work. Under review by the International Conference on Machine Learning (ICML). Do not distribute.

**Input**

Q: Evaluate $\log_{\sqrt{5}} 125\sqrt{5}$

### Few-shot Chain-of-Thought (Wei et al., 2022)

**Model Output**

A: We can write $125\sqrt{5}$ as $5^3 \cdot 5^{1/2} = 5^{7/2}$.
The logarithm base $\sqrt{5}$ of $5^{7/2}$ is the exponent we need to raise $\sqrt{5}$ to get $5^{7/2}$.
Since $\sqrt{5} = 5^{1/2}$, we have $(5^{1/2})^{2n} = 5^n$ for any integer $n$.
So, we need to find $n$ such that $5^n = 5^{7/2}$
Clearly, $n = 7/2$.

Therefore, $\log_{\sqrt{5}} 125\sqrt{5} = \boxed{\frac{7}{2}}$. 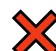

### Learning Principles (LEAP; this work)

**Generating Mistakes Intentionally (performed once)**

Q: Evaluate $\log_{1/3} 9$
A: To evaluate $\log_{1/3} 9$, you want to find the exponent to which $\frac{1}{3}$ must be raised to get 9 as the result.
...
$\frac{1^x}{(1/3)^x} = \frac{3^2}{1}$
So $\log_{1/3} 9 = \boxed{2.}$ 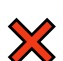

**Learning Principles (performed once)**

It is crucial to accurately calculate the powers of the base in logarithm problems to correctly determine the exponent. Additionally understanding the definition of a logarithm is key: the logarithm base b of a number x is the exponent to which b must be raised to get x.

**Inference with Learned Principles:**

A: First we can write $125\sqrt{5}$ as $5^3 \cdot 5^{1/2} = 5^{7/2}$.
The logarithm base $\sqrt{5}$ of $5^{7/2}$ is asking to which power we must raise $\sqrt{5}$ to get $5^{7/2}$.
Since $\sqrt{5} = 5^{1/2}$, we can see that $(5^{1/2})^7 = 5^{7/2}$.

Therefore, $\log_{\sqrt{5}} 125\sqrt{5} = \boxed{7.}$ 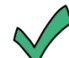

Figure 2: An illustration of LEAP: Given a few input-output examples, Chain-of-Thought (left) generates a response to the test question by directly learning from the (correct) examples. In contrast, Learning Principles (LEAP, right) first (a) generates *mistaken* zero-shot Chain-of-Thought response for each given input-output example by sampling with a non-zero temperature; (b) generates explicit *principles* by providing the LLM with the mistaken CoT along with the correct output; and finally (c) generates a response to the test question, by providing the LLM with both the given input-output examples *and the learned principles*. Note that steps (a) and (b) are performed once per task.

Chialvo & Bak, 1999; Edmondson, 1999). In fact, learning from mistakes is also a fundamental concept of machine learning, which goes back to classical work such as Wiener (1948); Rosenblatt (1957) and Minsky & Papert (1969), but is not utilized by current ICL and prompting methods.

Differently from machines, humans can often *verbalize* their mistakes and articulate explicit *principles*, or "lessons"; these principles can further help humans avoid these and similar mistakes in the future. Inspired by this ability and the benefits it provides, we propose Learning Principles

(LEAP): A prompting approach for learning principles from mistakes, and then conditioning on these principles when responding to *other* inputs. Instead of providing the model *only* with the given (correct) few-shot examples, LEAP begins by (a) inducing the LLM to *make mistakes* on each of these given few-shot inputs, in a zero-shot fashion, by sampling outputs with a non-zero temperature; (b) generating explicit *principles* by providing the same LLM with the mistaken outputs along with the correct output; and finally (c) generating a response to the test question as in standard few-shot ICL, while providing the LLM with both the given

input-output examples *and the learned principles*. LEAP is illustrated in Figure 2; examples for some of the learned principles are provided in Figure 1.

Crucially, LEAP does not require any more input than the standard few-shot prompting settings. Further, the principle learning step is applied *once* for a task, and then the same learned principles are used for the entire test set. If the few-shot examples are given as Chain-of-Thought (CoT; Wei et al., 2022c), LEAP is applied seamlessly by generating *mistaken* chains-of-thought, contrasting them with the correct chains-of-thought, and generating principles.

We demonstrate the effectiveness of LEAP across a wide range of reasoning benchmarks, including mathematical reasoning in GSM8K (Cobbe et al., 2021) and MATH (Hendrycks et al., 2021), multi-hop question answering tasks in HotpotQA (Yang et al., 2018b), textual reasoning in DROP (Dua et al., 2019a), and 27 Big-Bench Hard (Suzgun et al., 2022) tasks. LEAP outperforms the standard practice of few-shot prompting using strong models such as GPT-3.5-turbo, GPT-4, GPT-4-turbo and Claude-2.1, with and without Chain-of-Thought. For example, LEAP improves over the standard few-shot CoT using GPT-4 by 7.5% in DROP, by 3.3% in HotpotQA, and on 16 tasks in Big-Bench Hard. These results suggest that LEAP revolutionizes the "traditional" concept of few-shot ICL, by leveraging the recently emerged abilities of LLMs to follow instructions (Mishra et al., 2021; Wei et al., 2022a; Ouyang et al., 2022) and explain mistakes given the correct answer or feedback (Madaan et al., 2023; Chen et al., 2023).

## 2. Background: In-Context Learning

In-context learning, also known as few-shot prompting, uses a few (typically between 3 and 8) input-output task-specific examples for teaching a pre-trained LLM to solve a downstream task (Brown et al., 2020; Liu et al., 2021; 2023). These input-output pairs $\mathcal{P} = \{(x_i, y_i)\}_{i=1}^{k}$ are concatenated to form a prompt $p = \langle x_1 \cdot y_1 \rangle \oplus \langle x_2 \cdot y_2 \rangle \oplus \ldots \oplus \langle x_k \cdot y_k \rangle$, where "·" denotes the concatenation of each input with its corresponding output, and "⊕" denotes the sequential combination of these pairs. Then, every new test input $x_{\text{test}}$ is appended to this prompt, resulting in p $\oplus \langle x_{\text{test}}\cdot$, which is then provided to the LLM . The LLM completes this input, and generates the corresponding output $\hat{y}_{\text{test}}$.

Typically, the answer $y$ additionally contains an explanation (or *thought*) for deriving the answer (Wei et al., 2022c). In these settings, each solution $y$ contains a thought $t$ and the final answer $a$, forming: $y_i = t_i \cdot a_i$, and the LLM is expected to generate the test-thought before generating the final test-answer: $t_{test} \cdot a_{test}$. In this work, we focus on this *chain-of-thought* setup, because of its widely demonstrated

effectiveness for reasoning tasks (Wang et al., 2022a; Wei et al., 2022c; Zhou et al., 2022; Wang et al., 2022b).

## 3. LEAP: Learning Principles from Mistakes

In LEAP, our goal is to learn general *principles* that help the model avoid potential mistakes in a downstream task. Given the few-shot examples $\mathcal{P} = \{\langle x_i, y_i \rangle\}_{i=1}^{k}$ for a certain task, we start by generating *mistakes* to the few given examples.

**Generating Mistakes** For each input-output pair $\langle x_i, y_i \rangle \in \mathcal{P}$, we generate a diverse set of solutions in a zero-shot fashion. Specifically, we follow Kojima et al. (2022), and create a zero-shot chain-of-thought prompt using $x_i$ and the phrase *Think step-by-step*. For each input $x_i$, we sample $n = 15$ outputs with a non-zero temperature, producing a varied set of potential solutions $\{\hat{y}_i^j\}_{j=1}^{n}$ for each example $x_i$, such that $\hat{y}_i^j = \hat{t}_i^j \cdot \hat{a}_i^j$, where $\hat{t}_i^j$ represents the intermediate reasoning steps (thoughts), and $\hat{a}_i^j$ denotes the final answer.

We identify *incorrect* solutions by comparing each $\hat{a}_i^j$ with the ground-truth answer $a_i$ (which is given as part of the task), forming a set of mistakes for each $\langle x_i, y_i \rangle \in \mathcal{P}$ pair:
$$\mathcal{M}_i = \left\{ \left( x_i, y_i, \hat{y}_i^j \right) \right\}_{j=1}^{n'}, \text{ such that } \forall j : \hat{a}_i^j \neq a_i.$$

**Generating Low-Level Principles** Then, for each such mistake in $\mathcal{M}$, we prompt the LLM to generate a natural language explanation of the mistake. In this step, the LLM is provided with the ground truth answer $y_i$, to articulate the rationale behind the solution's inaccuracy, as illustrated in Figure 3. The insights of how to avoid these mistakes are aggregated across all examples to form a set of low-level principles, denoted as $\mathcal{L}_{\text{LOW-LEVEL}}$.

**Generating High-Level Principles** Subsequently, we use the LLM to condense the low-level principles into approximately 5 key bullet points, thus creating *high*-level principles, denoted as $\mathcal{L}_{\text{HIGH-LEVEL}}$. The motivation for this step is generating *generic*, example-agnostic, principles for solving the downstream task, that do not depend on mistakes made for any specific example.

**Final Inference on Unseen Examples** These principles, either low or high-level, are then appended to the prompt $p$ , forming enhanced prompts: $p_{\text{LOW-LEVEL}} = \mathcal{L}_{\text{LOW-LEVEL}} \oplus p$ for low-level feedback, and $p_{\text{HIGH-LEVEL}} = \mathcal{L}_{\text{HIGH-LEVEL}} \oplus p$ for high-level feedback, where $p$ is the prompt constructed using the standard concatenation of the few-shot examples, as described in Section 2.

Finally, we use the enhanced prompts $p_{\text{LOW-LEVEL}}$ or $p_{\text{HIGH-LEVEL}}$ for answering all unseen examples in the test set. In effect, the one-time process of generating principles

```
Question: {question}

Generated Reasoning: {response}

Generated Answer: {generated_answer}

Correct Reasoning: {correct_reasoning}

Correct Answer: {correct_answer}

Instruction: Conduct a thorough analysis of the generated answer in comparison to the
correct answer. Also observe how the generated reasoning differs from the correct
reasoning. Identify any discrepancies, misunderstandings, or errors. Provide clear
insights, principles, or guidelines that can be derived from this analysis to improve
future responses. We are not focused on this one data point, but rather on the general
principle.

Reasoning: <discuss why the generated answer is wrong>
Insights: <what principle should be looked at carefully to improve the performance in
the future>
```

Figure 3: LEAP prompt to help LLM evaluate its own generated reasoning and answers, contrasting them with the correct reasoning and answers. The LLM is prompted to identify errors in its reasoning and extract key insights for improvement. This figure specifically represents the 'GenerateExplanation' step in the LEAP algorithm (Algorithm 1).

helps learn and *articulate* insights from the model's previous mistakes, potentially improving its accuracy and reasoning capabilities in future responses for *other* inputs.

The complete algorithm is summarized in Algorithm 1. Although different LLMs can be used for each step, in all our experiments, we fixed the LLM across all steps: generating mistakes, generating principles from those mistakes, and testing using these self-generated principles. This ensures that the difference in results arises only from LEAP rather than any kind of model ensemble or teaching.

## 4. Evaluation

We evaluated LEAP across various reasoning tasks, including HotpotQA (Yang et al., 2018b), DROP (Dua et al., 2019a), MATH (Hendrycks et al., 2021), GSM8K (Cobbe et al., 2021), and Big-Bench Hard (Suzgun et al., 2022). We follow the standard few-shot Chain-of-Thought (CoT ); we apply LEAP on top of few-shot CoT using the exact same number of labeled examples. Importantly, principles are generated *once* for every LLM and benchmark.

### 4.1. Experiment Setup

**Compared Approaches**  Our baseline is the standard Few-shot prompting with CoT . We compare this baseline with two variants of our proposed approach LEAP:

- **LEAP** HIGH-LEVEL is our proposed approach, as described in Section 3.

- **LEAP** LOW-LEVEL is similar to LEAP HIGH-LEVEL, except that we skip the step of "Generating High-Level Principles", and test the models on the downstream task using the few-shot examples *and the low-level principles*, using $\mathcal{L}_{\text{LOW-LEVEL}}$ rather than $\mathcal{L}_{\text{HIGH-LEVEL}}$ (Section 3).

**Benchmarks**  We used diverse reasoning benchmarks:

- **Textual Reasoning**: HotpotQA (Yang et al., 2018a) is a question-answering dataset of computational questions that require multi-hop reasoning. DROP (Dua et al., 2019b) is a reading comprehension dataset that requires numerical and logical reasoning over textual paragraphs; for evaluation, we randomly sampled 2000 questions from its dev set.

- **Mathematical Reasoning**: GSM8K (Cobbe et al., 2021) comprises a test set of 1,319 diverse grade school math word problems, curated by human problem writers. In MATH (Hendrycks et al., 2021), there are 5,000 diverse examples consisting of problems from mathematics competitions. These are the two most common mathematical reasoning benchmarks.

- **Big-Bench Hard** (Suzgun et al., 2022): contains 27 challenging tasks that test various reasoning capabilities of LLMs . We repeated every run 3 times with a temperature of zero and report the average.[1]

---

[1]Internal non-determinism causes different outputs even with a temperature of zero, but the variance was negligible.

---

**Algorithm 1** LEAP Algorithm

---

**Require:** Few-shot examples $\mathcal{P} = \{\langle x_i, y_i \rangle\}_{i=1}^k$, a pretrained LLM , number of outputs per input $n$, high-temperature setting $T$

 0: **for** each input-output pair $\langle x_i, y_i \rangle$ in $\mathcal{P}$ **do**

 0:      $\mathcal{S}_i \leftarrow$ ZeroShotCoT $(LLM, x_i, n, T)$ {Generate solutions using zero-shot chain-of-thought prompting}

 0:      $\mathcal{M}_i \leftarrow \{(x_i, y_i, \hat{y}_i^j) \in \mathcal{S}_i : \hat{a}_i^j \neq a_i^j\}$ {Identify incorrect solutions}

 0:      **for** each $x_i, y_i, \hat{y}_i$ in $\mathcal{M}_i$ **do**

 0:          $\mathcal{L}_{\text{LOW-LEVEL},i} \leftarrow$ GenerateLowLevelPrinciples $(LLM, x_i, \hat{y}_i, y_i)$ {Generate principles for each mistake}

 0:      **end for**

 0: **end for**

 0: $\mathcal{L}_{\text{LOW-LEVEL}} \leftarrow \bigcup_{i=1}^k \mathcal{L}_{\text{LOW-LEVEL},i}$ {Aggregate low-level principles}

 0: $\mathcal{L}_{\text{HIGH-LEVEL}} \leftarrow$ GenerateHighLevelPrinciples $(LLM, \mathcal{L}_{\text{LOW-LEVEL}})$ {Generate high-level principles}

 0: $p_{\text{LOW-LEVEL}} \leftarrow$ Concatenate $(\mathcal{L}_{\text{LOW-LEVEL}}, \mathcal{P})$ {Create enhanced prompt with low-level principles}

 0: $p_{\text{HIGH-LEVEL}} \leftarrow$ Concatenate $(\mathcal{L}_{\text{HIGH-LEVEL}}, \mathcal{P})$ {Create enhanced prompt with high-level principles}

      **return** $p_{\text{LOW-LEVEL}}, p_{\text{HIGH-LEVEL}}$ =0

---

Table 1: **Textual Reasoning results**: Accuracy in textual reasoning benchmarks. The best approach for each base LLM in each dataset is in **bold**; the second-best approach is underlined. We see a good performance boost by adopting LEAP with high-level feedback and low-level feedback. Almost all the models can benefit from the principles learned and fix their previous mistakes.

|  |  | GPT-3.5-turbo | GPT-4 | GPT-4-turbo | Gemini Pro |
|---|---|---|---|---|---|
| HotpotQA | Few-shot CoT | 29.10 | 36.35 | 38.10 | **28.25** |
|  | LEAP LOW-LEVEL | **32.60** | 39.30 | 37.85 | 23.70 |
|  | LEAP HIGH-LEVEL | 30.35 | **39.65** | **38.75** | 25.50 |
| DROP | Few-shot CoT | 63.20 | 72.05 | 83.40 | 64.60 |
|  | LEAP LOW-LEVEL | **63.35** | **79.55** | **83.60** | 67.15 |
|  | LEAP HIGH-LEVEL | 63.00 | 78.60 | 80.00 | **67.60** |

**Models** We evaluated LEAP across a wide range of base models, including `GPT-3.5-turbo` (version `-0613`), `GPT-4` (version `-0613`), `GPT-4-turbo` (version `-1106`), `Claude-2.1`, and `Gemini Pro` (Gemini Team Google, 2023).

**Few-shot examples** In Big-Bench Hard, we used the CoT prompts from Suzgun et al. (2022) with three given examples (3-shot) in each benchmark. In MATH and GSM8K , we used the standard training examples from each dataset, with 3 examples for each. In DROP, we used 3 given examples that we adopted from Least-to-Most (Zhou et al., 2022). In HotpotQA we used 6 examples from ReAct (Yao et al., 2022) ("closed-book"). Importantly, in each benchmark, the exact same few-shot examples were used across all evaluated approaches, including the baseline and LEAP. Our complete prompts are provided in Appendix G.

### 4.2. Textual Reasoning Results

Table 1 shows the results on DROP and HotpotQA. As shown, LEAP improves over the Few-shot CoT baseline by up to 3.5% on Hotpot QA and 7.5% on DROP. In HotpotQA, GPT-3.5-Turbo and GPT-4 are consistently improved when

using LEAP. In DROP, GPT-4 is significantly improved by LEAP (an absolute gain of 7.5%), Gemini Pro is improved by 3%, while the improvement for GPT-3.5-Turbo and GPT-4-turbo are more minor.

In most tasks and base models, both LEAP LOW-LEVEL and LEAP HIGH-LEVEL improve over the Few-shot CoT baseline. The only case where the Few-shot CoT baseline performs better than both LEAP LOW-LEVEL and LEAP HIGH-LEVEL is in HotpotQA using Gemini Pro. Observing the low-level principles that Gemini Pro learned in HotpotQA (Table 41), we believe that the learned principles are correct and useful, but they are overly focused on the examples they were generated for, more verbose, and similar to each other. These hinder the principles (and the high-level principles generated from them) from generalizing to other examples. For *zero*-shot prompting using principles learned from few examples, additional results are shown in Appendix A.

### 4.3. Math Reasoning Results

Table 2 shows the results on MATH and GSM8K . As shown, in GPT-3.5-turbo and GPT-4, both LEAP LOW-LEVEL *and* LEAP HIGH-LEVEL outperform the Few-shot CoT baseline. Claude-2 shows inconclusive results: in GSM8K , Few-

Table 2: **Math Reasoning Results**: Accuracy in MATH and GSM8K . The best approach for each LLM and base task is in **bold**; the second-best approach is underlined. LEAP $_{\text{HIGH-LEVEL}}$ and LEAP $_{\text{LOW-LEVEL}}$ in both GSM8K and MATH datasets consistently improve the performance over the CoT baseline. We also observe that the learned mathmetical principles can be generalized to different test questions.

|  |  | GPT-3.5-turbo | GPT-4 | Claude-2 | Gemini Pro |
|---|---|---|---|---|---|
| GSM8K | Few-shot CoT | 76.4 | 93.6 | **84.3** | 77.8 |
|  | LEAP $_{\text{LOW-LEVEL}}$ | **77.4** | **94.2** | 82.7 | 77.3 |
|  | LEAP $_{\text{HIGH-LEVEL}}$ | 76.6 | 93.8 | 83.8 | **78.7** |
| MATH | Few-shot CoT | 55.6 | 63.5 | 43.2 | **31.1** |
|  | LEAP $_{\text{LOW-LEVEL}}$ | 56.1 | **64.5** | 42.7 | 29.7 |
|  | LEAP $_{\text{HIGH-LEVEL}}$ | **56.5** | 64.0 | **43.4** | 30.3 |

Table 3: **Llama-2-chat-70B Results**: Accuracy in MATH and GSM8K using Llama-2-chat-70B as the base LLM, while generating the principles either with Llama-2-chat-70B or GPT-4. Even when the principles are generated by GPT-4 (GPT-4 Critic), Llama-2-chat-70B does not leverage the learned principles, and does not improve over the baseline Few-shot CoT.

| Llama-2-chat-70B as the base model, with: |  | Llama-2 Critic | GPT-4 Critic |
|---|---|---|---|
| GSM8K | Few-shot CoT | **52.5** | **52.5** |
|  | LEAP $_{\text{LOW-LEVEL}}$ | 50.6 | 49.8 |
|  | LEAP $_{\text{HIGH-LEVEL}}$ | 47.0 | 51.0 |
| MATH | Few-shot CoT | **16.2** | **16.2** |
|  | LEAP $_{\text{LOW-LEVEL}}$ | 13.8 | 12.9 |
|  | LEAP $_{\text{HIGH-LEVEL}}$ | 14.2 | 13.8 |

shot CoT performs better than LEAP; in MATH, LEAP $_{\text{HIGH-LEVEL}}$ achieves slightly higher accuracy than Few-shot CoT . Gemini Pro also shows inconclusive results, where both LEAP $_{\text{LOW-LEVEL}}$ and LEAP $_{\text{HIGH-LEVEL}}$ outperform the baseline in GSM8K , but perform slightly worse in MATH. Additional results, for zero-shot prompting using principles learned from few examples, are shown in Appendix B.

**Does leap work with open-source models?** In our preliminary experiments with open-source models, LEAP did not improve over the few-shot CoT baseline. While the open-source models *did* produce useful principles, the open-source models did not follow these principles at test time.

Table 3 shows some of these experiments with Llama-2-chat-70B : we used Llama-2-chat-70B as the base model, but generated the principles using either Llama-2-chat-70B ("LLama-2 Critic") or with GPT-4 ("GPT-4 Critic"). As shown, even when the principles are generated by GPT-4 (which we assume to be useful), Llama-2-chat-70B does not manage to leverage them to generate better final responses with either LEAP $_{\text{LOW-LEVEL}}$ or LEAP $_{\text{HIGH-LEVEL}}$ .

In general, we believe that LEAP requires a base LLM with strong enough instruction following and reflection capabilities; we believe that, unfortunately, most open-source models are not as powerful as proprietary models yet.

### 4.4. Big-Bench Hard Results

Figure 4 shows the results on selected BBH tasks; results for the rest of the BBH tasks are shown in Table 7 in Appendix C. We selected tasks to Table 6 where the difference in results was the most meaningful: there were some tasks where GPT-4-0613 achieved 100% accuracy with *any* of the approaches, including both LEAP and the baseline.

As shown in Table 6, in 37 out of 42 combinations of task and LLM , one of LEAP $_{\text{LOW-LEVEL}}$ or LEAP $_{\text{HIGH-LEVEL}}$ outperforms the baseline Few-shot CoT . In 24 of the cases, both LEAP $_{\text{LOW-LEVEL}}$ *and* LEAP $_{\text{HIGH-LEVEL}}$ outperform the Few-shot CoT baseline. In two cases, the Few-shot CoT baseline performs equally to LEAP $_{\text{LOW-LEVEL}}$ ; and in 3 cases the Few-shot baseline performs better than both LEAP $_{\text{LOW-LEVEL}}$ and LEAP $_{\text{HIGH-LEVEL}}$ .

We could not identify any particular pattern as to which method should be used: LEAP $_{\text{LOW-LEVEL}}$ or LEAP $_{\text{HIGH-LEVEL}}$ ; it seems that this depends on the reasoning complexity of the task, the diversity across questions in the benchmark, and the quality of the principles that the LLM had managed to generate from the 3 given examples for that task. We thus suggest that in real-life scenarios, both approaches should be tested, and selected using a validation set. Additional results, for zero-shot prompting using principles learned

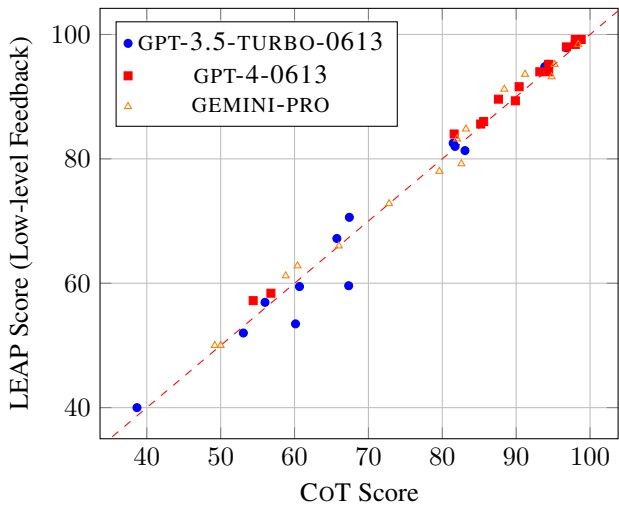 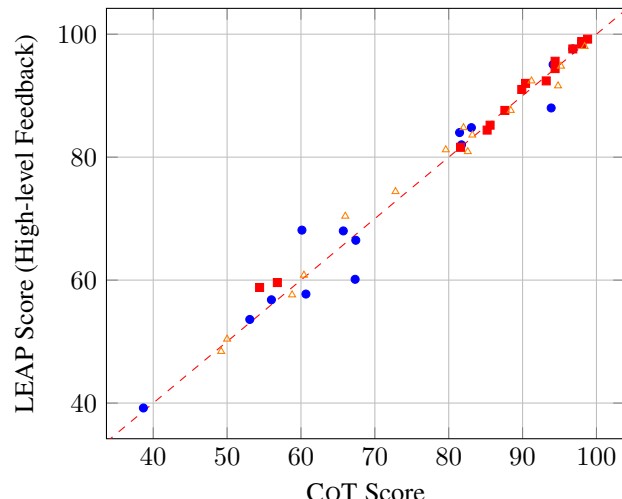

Figure 4: Accuracy in BBH tasks, across GPT-3.5-TURBO-0613 , GPT-4-0613 , and GEMINI-PRO . The figure presents the results using a scatter plot, where the y-axis represents scores achieved with LEAP, and the x-axis represents the baseline scores from CoT . Each task is represented by a point on the plot, with different shapes assigned to different models for easy distinction. Tasks above the $y = x$ line are those where LEAP leads to an improvement in performance. Table 6 shows the detailed results for all 27 Big-Bench hard tasks. We find that in 37 out of 42 combinations of task and LLM , one of LEAP LOW-LEVEL or LEAP HIGH-LEVEL outperforms the baseline Few-shot CoT .

from a few examples, are shown in Appendix C.

## 5. Qualitative Analysis

Figure 5 shows examples of questions from BBH , along with the task's learned principles, the baseline few-shot CoT response, and the LEAP LOW-LEVEL response, all generated by GPT-3.5-turbo. The learned principle in each case is highlighted, along with the mistake in the CoT output and the correct reasoning generated by LEAP.

Additional examples are shown in Appendix D. The actual learned principles for some of the tasks and LLMs are provided in Tables 9-43 in Appendix E.

## 6. Related Works

This section focuses on the few works most related to ours. For a complete survey of relevant papers, see Zhao et al. (2023); Qiao et al. (2023); Kaddour et al. (2023); Xi et al. (2023); Zhang et al. (2023).

**Comparison to Madaan et al. (2023)** A related work to ours is Self-Refine (Madaan et al., 2023), which, similarly to LEAP, uses the LLM to reflect on its own previously generated outputs. However, a significant difference from Self-Refine is that applying Self-Refine to a test example requires feedback *for that specific test example*; contrarily, in this work, we show that principles that were learned from

some examples *generalize to other* test examples which we do not have feedback for. Further, Self-Refine uses "intrinsic self-correction", without any external feedback; as found by recent work (Huang et al., 2023a; Valmeekam et al., 2023), such intrinsic self-correction does not benefit *reasoning* tasks. In contrast, in this work, we show that the external signal provided by as few as 3 examples is sufficient for effective self-reflection in reasoning if these are used to articulate explicit learned principles. Another related work is MemPrompt (Madaan et al., 2022), which stores feedback on mistakes in a memory, and retrieves relevant feedback for a new query; in contrast, LEAP is a one-time process that does not require continuous storage and retrieval.

**Comparison to Yang et al. (2023)** Optimization by PROmpting (OPRO) (Yang et al., 2023) uses LLMs to improve zero-shot prompts, and both OPRO and LEAP can be thought of as ways of improving a prompt for a specific task. However, applying OPRO for prompt optimization requires a *training set*, which can be as large as ∼**260** examples for GSM8K; in contrast, LEAP requires only **3** labeled examples. Further, OPRO calls the LLM for each of these ∼**260** examples in each step; while the number of steps can be as low as 6, it can also be as high as 107. This results in between ∼**1500** calls and up to $107 \times 260 \approx 27{,}000$ inference calls to the LLM. In contrast, LEAP uses only about **34** inference calls: 3 calls to generate mistakes; one call for each mistake to generate low-level principles, which results in about 30 calls; and a final call to generate high-level

---

**Boolean Expressions (BBH)**

**Question:** not not ( False and not False ) is
**Principles:** .. Double negation, as in 'not not', cancels out and returns the original value. ...
**Few-shot CoT:** Let's think step by step. Remember that (i) expressions inside brackets are always evaluated first and that (ii) the order of operations from highest priority to lowest priority is "not", "and", "or", respectively. We first simplify this expression "Z" as follows: "Z = not not ( False and not False ) = ... = not not (False) = not not True = not False = True. So the answer is $\boxed{True}$ . ✗
**LEAP** LOW-LEVEL **Response:** Let's think step by step. Remember that (i) expressions inside brackets are always evaluated first and that (ii) the order of operations from highest priority to lowest priority is "not", "and", "or", respectively. We first simplify this expression "Z" as follows: "Z = not not ( False and not False ) = ... = not not (False) = not not False = False. So the answer is $\boxed{False}$ . ✓

---

**Object Counting (BBH)**

**Question:** I have a flute, a piano, a trombone, four stoves, a violin, an accordion, a clarinet, a drum, two lamps, and a trumpet. How many musical instruments do I have?
**Task Principles:** The response should be directly related to the user's query or statement. Irrelevant information can confuse the user and detract from the overall quality of the response...
**Few-shot CoT:** Let's think step by step. We first identify the musical instruments on the list and include their quantity in parentheses: - flute (1) - piano (1) - trombone (1) - stoves (4) - violin (1) - accordion (1) - clarinet (1) - drum (1) - lamps (2) - trumpet (1) Now, let's add the numbers in parentheses: 1 + 1 + 1 + 4 + 1 + 1 + 1 + 1 + 2 + 1 = 14. So the answer is $\boxed{14}$ . ✗
**LEAP** LOW-LEVEL **Response:** Let's think step by step. We first identify the musical instruments on the list and include their quantity in parentheses: - flute (1) - piano (1) - trombone (1) - violin (1) - accordion (1) - clarinet (1) - drum (1) - trumpet (1) Now, let's add the numbers in parentheses: 1 + 1 + 1 + 1 + 1 + 1 + 1 + 1 = 8. So the answer is $\boxed{8}$ . ✓

---

Figure 5: Examples from the Boolean Expressions (left) and Object counting (right) tasks from BBH . The learned principle is highlighted in yellow, the mistaken step of the baseline is highlighted in red, and the correct use of the principle by LEAP is highlighted in green. This demonstrates how the learned principles guide LEAP in generating a better answer.

principles. Moreover, like OPRO, LEAP can also improve zero-shot prompting by learning principles from a few examples and applying them in a zero-shot fashion, as we show in Appendix A, Appendix B, and Appendix C. For example, Zero-shot-LEAP improves the strong GPT-4 model by 1% on GSM8K (Table 5). Similarly, EvoPrompt (Guo et al., 2023) requires around **10,000** LLM inference calls, which is also several orders of magnitude more than LEAP. Other related approaches are AutoPrompt (Shin et al., 2020), STaR (Zelikman et al., 2022), LMSI (Huang et al., 2023b), and Self-Align (Sun et al., 2023) but these are based on *training* the LLM , which is often either inaccessible or computationally infeasible.

**Comparison to Chia et al. (2023)**   Contrastive Chain-of-Thought (Chia et al., 2023) shares a similar motivation with our work: learning from negative in-context examples instead of learning only from positive examples. However, when we reproduced their results, we found that simple improvements to the post-processing code of extracting the final answer out of the LLM's raw output led to differences of more than 10 absolute points from their reported results in GSM8K , which made the proposed Contrastive CoT approach perform similarly or worse than the baseline few-shot CoT . We thus believe that generating *explicit* principles, as in LEAP, is the key ingredient in learning from negative in-context examples.

## 7. Conclusion

In this paper, we introduce Learning Principles (LEAP), a novel approach that allows LLMs to learn *more* out of given few-shot examples, by intentionally making mistakes on these examples; reflecting on the mistakes; and finally articulating explicit task-specific principles, which helps avoid similar mistakes in the future. LEAP requires exactly the same number of labeled examples as few-shot prompting, and allows improving a variety of strong LLMs (GPT-3.5-turbo, GPT-4, GPT-4-turbo and Gemini Pro) across a broad range of reasoning tasks (DROP, HotpotQA, GSM8K , MATH, and Big-Bench Hard). We believe that LEAP unlocks new possibilities from learning in the traditional concept of few-shot in-context learning, by learning from *mistakes*, rather than learning from positive examples only.

## 8. Broader Impact

This paper aims to advance the field of Machine Learning by presenting a method that allows LLMs to learn from their own mistakes. The potential of LLMs to learn to correct their behaviors from past mistakes can significantly improve their capability without human-in-the-loop. This ability has many potential societal consequences, most notably the opportunity for LLMs to self-improve across a wide range of tasks where their initial performance may be lacking

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

# A. Additional Results on Text Reasoning

Table 4 shows additional Text Reasoning results. The upper part of Table 4 is identical to Table 1; the lower part of Table 4 contains zero-shot results, with principles that were learned from the given few-shot examples; this setup is similar to related work such as (Yang et al., 2023).

Table 4: **Text Reasoning Results**: Accuracy in textual reasoning datasets. The best approach for each base LLM in each dataset is in **bold**; the second-best approach is underlined.

| | | GPT-3.5-turbo | GPT-4 | Gemini Pro |
|---|---|---|---|---|
| **Few-shot results, identical to Table 2:** | | | | |
| HotpotQA | Few-shot CoT | 29.10 | 36.35 | **28.25** |
| | LEAP LOW-LEVEL | **32.60** | 39.30 | 23.70 |
| | LEAP HIGH-LEVEL | 30.35 | **39.65** | 25.50 |
| DROP | Few-shot CoT | 63.20 | 72.05 | 64.60 |
| | LEAP LOW-LEVEL | **63.35** | **79.55** | 67.15 |
| | LEAP HIGH-LEVEL | 63.00 | 78.60 | **67.60** |
| | | GPT-3.5-turbo | GPT-4 | Gemini Pro |
| **Zero-shot results, using principles learned from few-shot examples:** | | | | |
| HotpotQA | Zero-shot CoT | 12.55 | 27.80 | **19.75** |
| | LEAP LOW-LEVEL | 6.65 | **31.20** | 3.30 |
| | LEAP HIGH-LEVEL | **13.10** | 30.90 | 2.45 |
| DROP | Zero-shot CoT | **59.25** | 82.20 | 62.75 |
| | LEAP LOW-LEVEL | 55.20 | **84.10** | **65.30** |
| | LEAP HIGH-LEVEL | 55.25 | 81.80 | 62.80 |

# B. Additional Results on Mathematical Reasoning

Table 5: **Math Reasoning Results**: Accuracy in MATH and GSM8K . The best approach for each LLM and base task is in **bold**; the second-best approach is underlined.

| | | GPT-3.5-turbo | GPT-4 | Claude-2 | Gemini Pro |
|---|---|---|---|---|---|
| **Few-shot results, identical to Table 2:** | | | | | |
| GSM8K | Few-shot CoT | 76.4 | 93.6 | **84.3** | |
| | LEAP LOW-LEVEL | **77.4** | **94.1** | 82.7 | |
| | LEAP HIGH-LEVEL | 76.6 | 93.8 | 83.8 | |
| MATH | Few-shot CoT | 55.6 | 63.5 | 43.2 | |
| | LEAP LOW-LEVEL | 56.1 | **64.5** | 42.7 | |
| | LEAP HIGH-LEVEL | **56.5** | 64.0 | **43.4** | |
| | | GPT-3.5-turbo | GPT-4 | Claude-2 | Gemini Pro |
| **Zero-shot results, using principles learned from few-shot examples:** | | | | | |
| GSM8K | Zero-shot CoT | **76.9** | 93.2 | 75.4 | |
| | LEAP LOW-LEVEL | 74.4 | **94.2** | 76.7 | |
| | LEAP HIGH-LEVEL | 73.8 | 94.1 | **76.9** | |
| MATH | Zero-shot CoT | **54.2** | **63.5** | 40.2 | |
| | LEAP LOW-LEVEL | 52.0 | 63.2 | 40.5 | |
| | LEAP HIGH-LEVEL | 50.0 | 61.5 | **41.8** | |

Table 5 shows additional Mathematical Reasoning results. The upper part of Table 5 is identical to Table 2; the lower part of Table 5 contains zero-shot results, with principles that were learned from the given few-shot examples; this setup is similar to related work such as (Yang et al., 2023).

# C. Additional Results on BBH

Table 7 shows results on additional BBH tasks that we could not fit into Table 6. We selected tasks to Table 6 in the main paper where the difference was more meaningful. As shown in Table 7, there were some tasks such as `temporal_sequences`, `web_of_lies` and `tracking_shuffled_objects_five_objects` where GPT-4 achieved 100% accuracy with *any* of the approaches, including both LEAP and the few-shot baseline.

| task | wrong to correct | correct to wrong | wrong to wrong | correct to correct |
|---|---|---|---|---|
| temporal_sequences | 31.0 | 10.0 | 66.0 | 143.0 |
| snarks | 18.0 | 11.0 | 39.0 | 110.0 |
| disambiguation_qa | 10.0 | 8.0 | 76.0 | 156.0 |
| logical_deduction_seven_objects | 27.0 | 21.0 | 129.0 | 73.0 |
| object_counting | 4.0 | 1.0 | 4.0 | 241.0 |
| movie_recommendation | 14.0 | 11.0 | 33.0 | 192.0 |
| navigate | 2.0 | 1.0 | 12.0 | 235.0 |
| formal_fallacies | 14.0 | 9.0 | 99.0 | 128.0 |
| sports_understanding | 5.0 | 4.0 | 10.0 | 231.0 |
| boolean_expressions | 5.0 | 2.0 | 10.0 | 233.0 |
| web_of_lies | 2.0 | 1.0 | 1.0 | 246.0 |
| multistep_arithmetic_two | 12.0 | 8.0 | 33.0 | 197.0 |
| causal_judgement | 18.0 | 18.0 | 51.0 | 100.0 |
| salient_translation_error_detection | 9.0 | 16.0 | 102.0 | 123.0 |
| tracking_shuffled_objects_three_objects | 12.0 | 15.0 | 30.0 | 193.0 |
| word_sorting | 11.0 | 13.0 | 109.0 | 117.0 |
| logical_deduction_five_objects | 24.0 | 27.0 | 78.0 | 121.0 |
| tracking_shuffled_objects_five_objects | 12.0 | 16.0 | 56.0 | 166.0 |
| hyperbaton | 4.0 | 19.0 | 40.0 | 187.0 |
| logical_deduction_three_objects | 9.0 | 16.0 | 20.0 | 205.0 |
| tracking_shuffled_objects_seven_objects | 10.0 | 14.0 | 79.0 | 147.0 |
| dyck_languages | 0.0 | 0.0 | 0.0 | 250.0 |
| date_understanding | 3.0 | 9.0 | 31.0 | 207.0 |
| penguins_in_a_table | 13.0 | 13.0 | 20.0 | 100.0 |
| reasoning_about_colored_objects | 11.0 | 19.0 | 33.0 | 187.0 |
| ruin_names | 14.0 | 38.0 | 87.0 | 111.0 |
| geometric_shapes | 9.0 | 25.0 | 74.0 | 142.0 |

Table 8: Efficacy of LEAP Methods Across Various Tasks. This table provides an overview of the effectiveness of LEAP in modifying the correctness of responses across a range of tasks. It highlights the number of instances where answers changed from wrong to correct, correct to wrong, and the stability of responses (both correct and incorrect). Notable gains in tasks like 'temporal_sequences' and 'snarks' suggest significant improvements, whereas tasks like 'ruin_names' and 'geometric_shapes' show areas needing further methodological refinement.

Table 6: BBH **Results:** Accuracy in BBH tasks, across GPT-3.5-turbo, GPT-4, and Gemini Pro. The best approach for each base LLM in each tasks is in **bold**; the second-best approach is underlined. Each number represents the average across 3 identical runs with a temperature of zero.

| Task | Approach | GPT-3.5-turbo | GPT-4 | Gemini Pro |
|---|---|---|---|---|
| boolean_expressions | Few-shot CoT | 94.13 | 96.80 | 91.20 |
| | LEAP LOW-LEVEL | 94.93 | **98.00** | **93.60** |
| | LEAP HIGH-LEVEL | **95.07** | 97.60 | 92.40 |
| disambiguation_qa | Few-shot CoT | 65.73 | 85.60 | 66.00 |
| | LEAP LOW-LEVEL | 67.20 | **86.00** | 66.00 |
| | LEAP HIGH-LEVEL | **68.00** | 85.20 | **70.40** |
| formal_fallacies | Few-shot CoT | 56.00 | 81.60 | 58.80 |
| | LEAP LOW-LEVEL | **56.93** | **84.00** | **61.20** |
| | LEAP HIGH-LEVEL | 56.80 | 81.60 | 57.60 |
| hyperbaton | Few-shot CoT | 83.07 | 98.00 | 88.40 |
| | LEAP LOW-LEVEL | 81.33 | **99.20** | **91.20** |
| | LEAP HIGH-LEVEL | **84.80** | 98.80 | 87.60 |
| logical_deduction_five_objects | Few-shot CoT | **60.67** | 85.20 | 60.40 |
| | LEAP LOW-LEVEL | 59.47 | **85.60** | **62.80** |
| | LEAP HIGH-LEVEL | 57.73 | 84.40 | 60.80 |
| logical_deduction_seven_objects | Few-shot CoT | 38.67 | 56.80 | 49.20 |
| | LEAP LOW-LEVEL | **40.00** | 58.40 | **50.00** |
| | LEAP HIGH-LEVEL | 39.20 | **59.60** | 48.40 |
| movie_recommendation | Few-shot CoT | 81.47 | 90.40 | 83.20 |
| | LEAP LOW-LEVEL | 82.53 | 91.60 | **84.80** |
| | LEAP HIGH-LEVEL | **84.00** | **92.00** | 83.60 |
| multistep_arithmetic_two | Few-shot CoT | 81.73 | 93.20 | 79.60 |
| | LEAP LOW-LEVEL | **82.00** | **94.00** | 78.00 |
| | LEAP HIGH-LEVEL | **82.00** | 92.40 | **81.20** |
| navigate | Few-shot CoT | 94.27 | 98.00 | **95.20** |
| | LEAP LOW-LEVEL | **95.20** | **98.40** | **95.20** |
| | LEAP HIGH-LEVEL | 94.93 | **98.40** | 94.80 |
| object_counting | Few-shot CoT | 96.80 | 98.80 | **94.80** |
| | LEAP LOW-LEVEL | **97.87** | **99.20** | 93.20 |
| | LEAP HIGH-LEVEL | 97.60 | **99.20** | 91.60 |
| ruin_names | Few-shot CoT | 60.13 | 87.60 | 72.80 |
| | LEAP LOW-LEVEL | 53.47 | **89.60** | 72.80 |
| | LEAP HIGH-LEVEL | **68.13** | 87.60 | **74.40** |
| snarks | Few-shot CoT | 67.42 | 89.89 | **82.58** |
| | LEAP LOW-LEVEL | **70.60** | 89.33 | 79.21 |
| | LEAP HIGH-LEVEL | 66.48 | **91.01** | 80.90 |
| sports_understanding | Few-shot CoT | 93.87 | 94.40 | **98.40** |
| | LEAP LOW-LEVEL | **94.80** | **95.20** | **98.40** |
| | LEAP HIGH-LEVEL | 88.00 | 94.40 | 98.00 |
| word_sorting | Few-shot CoT | 53.07 | 94.40 | 82.00 |
| | LEAP LOW-LEVEL | 52.00 | 94.00 | 83.20 |
| | LEAP HIGH-LEVEL | **53.60** | **95.60** | **84.80** |

Table 7: Additional results on Big-Bench Hard tasks. The best approach for each base LLM in each tasks is in **bold**; the second-best approach is underlined. Each number represents the average across 3 identical runs with a temperature of zero.

| Task | Approach | GPT-3.5-turbo | GPT-4 | Gemini-Pro |
|---|---|---|---|---|
| geometric_shapes | Few-shot | **67.33** | 54.40 | 50.00 |
| | + LEAP LOW-LEVEL | 59.60 | 57.20 | 50.00 |
| | + LEAP HIGH-LEVEL | 60.13 | **58.80** | **50.40** |
| causal_judgement | Few-shot | **63.64** | **73.26** | **62.57** |
| | + LEAP LOW-LEVEL | **63.64** | 72.73 | **62.57** |
| | + LEAP HIGH-LEVEL | 63.10 | 72.73 | **62.57** |
| date_understanding | Few-shot | **86.67** | 90.00 | **87.60** |
| | + LEAP LOW-LEVEL | 83.73 | **91.20** | **87.60** |
| | + LEAP HIGH-LEVEL | 85.60 | 90.00 | 86.40 |
| dyck_languages | Few-shot | **35.73** | **56.80** | **0** |
| | + LEAP LOW-LEVEL | 33.07 | **56.80** | **0** |
| | + LEAP HIGH-LEVEL | **35.73** | 54.00 | **0** |
| logical_deduction _three_objects | Few-shot | **88.80** | **99.20** | 89.20 |
| | + LEAP LOW-LEVEL | 86.80 | 98.80 | 90.00 |
| | + LEAP HIGH-LEVEL | 85.73 | 98.80 | **90.80** |
| penguins_in_a_table | Few-shot | **76.94** | **97.26** | **81.51** |
| | + LEAP LOW-LEVEL | 73.74 | 96.58 | 78.77 |
| | + LEAP HIGH-LEVEL | 73.97 | 96.58 | 76.71 |
| reasoning_about _colored_objects | Few-shot | **82.13** | **95.20** | 83.20 |
| | + LEAP LOW-LEVEL | 77.87 | 91.20 | **84.00** |
| | + LEAP HIGH-LEVEL | 75.87 | 94.00 | 82.40 |
| salient_translation _error_detection | Few-shot | **55.73** | **68.80** | **56.00** |
| | + LEAP LOW-LEVEL | 55.20 | 67.20 | 52.80 |
| | + LEAP HIGH-LEVEL | 54.27 | 67.20 | 55.20 |
| temporal_sequences | Few-shot | 60.93 | **100.0** | **99.60** |
| | + LEAP LOW-LEVEL | **65.20** | **100.0** | 98.00 |
| | + LEAP HIGH-LEVEL | 62.80 | **100.0** | 99.20 |
| tracking_shuffled_objects _five_objects | Few-shot | **72.40** | **100.0** | 64.00 |
| | + LEAP LOW-LEVEL | 70.93 | **100.0** | **64.40** |
| | + LEAP HIGH-LEVEL | 71.33 | **100.0** | 64.00 |
| tracking_shuffled_objects _seven_objects | Few-shot | **65.07** | **100.0** | 56.8 |
| | + LEAP LOW-LEVEL | 62.53 | **100.0** | 60.0 |
| | + LEAP HIGH-LEVEL | 61.73 | 98.8 | **60.8** |
| tracking_shuffled_objects _three_objects | Few-shot | 83.60 | **100.0** | **69.60** |
| | + LEAP LOW-LEVEL | 82.80 | **100.0** | 69.20 |
| | + LEAP HIGH-LEVEL | **85.87** | **100.0** | 68.40 |
| web_of_lies | Few-shot | 98.67 | **100.0** | **100.0** |
| | + LEAP LOW-LEVEL | **99.20** | **100.0** | **100.0** |
| | + LEAP HIGH-LEVEL | **99.20** | **100.0** | **100.0** |

# D. Additional Qualitative Analysis

---

### Object Counting

**Question:** I have a flute, a piano, a trombone, four stoves, a violin, an accordion, a clarinet, a drum, two lamps, and a trumpet. How many musical instruments do I have?

**Principles:** Focus on relevance: The response should be directly related to the user's query or statement. Irrelevant information can confuse the user and detract from the overall quality of the response. ...

**CoT Response:** Let's think step by step. We first identify the musical instruments on the list and include their quantity in parentheses: - flute (1) - piano (1) - trombone (1) - stoves (4) - violin (1) - accordion (1) - clarinet (1) - drum (1) - lamps (2) - trumpet (1) Now, let's add the numbers in parentheses: $1 + 1 + 1 + 4 + 1 + 1 + 1 + 2 + 1 = 14$. So the answer is $\boxed{14}$. ✗

**LEAP** ₗₒw-ₗₑᵥₑₗ **Response:** Let's think step by step. We first identify the musical instruments on the list and include their quantity in parentheses: - flute (1) - piano (1) - trombone (1) - violin (1) - accordion (1) - clarinet (1) - drum (1) - trumpet (1) Now, let's add the numbers in parentheses: $1 + 1 + 1 + 1 + 1 + 1 + 1 + 1 = 8$. So the answer is $\boxed{8}$. ✓

---

### Salient Translation Error Detection

**Question:** The following translations from German to English contain a particular error. That error will be one of the following types: Named Entities, Numerical Values, Modifiers or Adjectives, Negation or Antonyms, Facts, Dropped Content. Please identify that error. Source: Neu Fahrland ist ein Ortsteil von Potsdam. Translation: Neu Fahrland is a district of Pots. The translation contains an error pertaining to Options: (A) Modifiers or Adjectives (B) Numerical Values (C) Negation or Antonyms (D) Named Entities (E) Dropped Content (F) Facts

**Principles:** When analyzing translations, it is important to carefully consider the translation of named entities. Any changes or inaccuracies in the translation of names, places, or other entities should be identified as named entity errors.

**CoT Response:** Let's think step by step. We solve this question by first translating the source sentence to English and then by comparing our translation with the provided translation. According to Google Translate, the correct translation of the source sentence from German to English is "Neu Fahrland is a district of Potsdam." On the other hand, the provided translation is "Neu Fahrland is a district of Pots." Note that the provided translation is missing the word "dam" at the end of Pots. Because a significant part of the translation is dropped, the translation contains an error pertaining to Dropped Content. So the answer is $\boxed{(E)}$. ✗

**LEAP** ₗₒw-ₗₑᵥₑₗ **Response:** Let's think step by step. We solve this question by first translating the source sentence to English and then by comparing our translation with the provided translation. According to Google Translate, the correct translation of the source sentence from German to English is "Neu Fahrland is a district of Potsdam." On the other hand, the provided translation is "Neu Fahrland is a district of Pots." Note that the provided translation omits the word "dam" from Potsdam. Because a significant part of the entity name is removed in the translation, the translation contains an error pertaining to Named Entities. So the answer is $\boxed{(D)}$. ✓

---

## Boolean Expressions

**Question:** not not ( False and not False ) is

**Principles:** ... 3. Double negation, as in 'not not', cancels out and returns the original value. ...

**CoT Response:** Let's think step by step. Remember that (i) expressions inside brackets are always evaluated first and that (ii) the order of operations from highest priority to lowest priority is "not", "and", "or", respectively. We first simplify this expression "Z" as follows: "Z = not not ( False and not False ) = not not ( A )" where "A = False and not False". Let's evaluate A: A = False and not False = False and True = False. Plugging in A, we get: Z = not not ( A ) = not not (False) = not not True = not False = True. So the answer is $\boxed{True}$. ✗

**LEAP** $_{\text{LOW-LEVEL}}$ **Response:** Let's think step by step. Remember that (i) expressions inside brackets are always evaluated first and that (ii) the order of operations from highest priority to lowest priority is "not", "and", "or", respectively. We first simplify this expression "Z" as follows: "Z = not not ( False and not False ) = not not ( A )" where "A = False and not False". Let's evaluate A: A = False and not False = False and True = False. Plugging in A, we get: Z = not not ( A ) = not not (False) = not not False = False. So the answer is $\boxed{False}$. ✓

## Snarks

**Question:** Which statement is sarcastic? Options: (A) It's not our job to fund education! It should be the government paying the maintenance to keep schools open (B) It's not our job to fund education! It should be the kids paying the maintenance to keep schools open

**Principles:** To improve future responses, it is important to carefully consider the context and tone of the statements when identifying sarcasm. Sarcasm often involves saying the opposite of what is meant ... Paying attention to the incongruity between the literal meaning of the words and the intended meaning can help in accurately identifying sarcasm.

**CoT Response:** Let's think step by step. If we look at (A), it suggests that funding education is not the responsibility of the speaker or the people, but rather the government. This statement is often used sarcastically to criticize individuals or groups who do not prioritize or support funding for education. If we look at (B), it suggests that funding education is not the responsibility of the speaker or the people, but rather the kids. This statement is not typically used sarcastically and can be seen as a genuine belief or opinion. Above the above, the sarcastic option is (A). So the answer is $\boxed{(A)}$. ✗

**LEAP** $_{\text{LOW-LEVEL}}$ **Response:** Let's think step by step. If we look at (A), it suggests that it is not the responsibility of the speaker or their group to fund education, and instead, it should be the government's responsibility. This statement aligns with the commonly accepted understanding that education funding is a shared responsibility between individuals, communities, and the government. Therefore, (A) does not appear to contain sarcasm. If we look at (B), it suggests that it is not the responsibility of the speaker or their group to fund education, and instead, it should be the kids' responsibility. This statement contradicts the generally accepted understanding that children should not be burdened with the financial responsibility of maintaining schools. The presence of such a contradiction in (B) suggests that it contains a taste of irony and sarcasm. Above the above, the sarcastic option is (B). So the answer is $\boxed{(B)}$. ✓

## Date Understanding

**Question:** May 6, 1992 is like yesterday to Jane, but that is actually ten years ago. What is the date one week ago from today in MM/DD/YYYY? Options: (A) 04/29/2002 (B) 04/24/2002 (C) 04/19/2002 (D) 04/28/2002 (E) 02/13/2002 (F) 05/20/2002

**Principles:** ... When subtracting days from a given date, it is important to accurately calculate the number of days to subtract and perform the subtraction correctly. ...

**CoT Response:** Let's think step by step. If May 6, 1992 is like yesterday to Jane, but that is actually ten years ago, then today's date is May 16, 2002. One week ago from today is May 9, 2002. So the answer is $(D)$. ✗

**LEAP** $_{\text{LOW-LEVEL}}$ **Response:** Let's think step by step. If May 6, 1992 is ten years ago, then today's date is May 6, 2002. One week ago from today is April 29, 2002, that is 04/29/2002. So the answer is $(A)$. ✓

## Multistep Arithmetic Two

**Question:** ((5 - 6 - 7 + 6) - (-1 - 1 * -7 + 9)) =

**Principles:** It is crucial to pay attention to the signs and operations when simplifying expressions inside parentheses. Any mistake in this step can lead to incorrect results.

**CoT Response:** Let's think step by step. Let's recall that the order of operations in mathematics is as follows: (1) Parentheses, (2) exponents, (3) multiplication and division (from left to right), (4) addition and multiplication (from left to right). So, remember to always compute the expressions inside parentheses or brackets first. This equation can be written as "A - B", where A = (5 - 6 - 7 + 6) and B = (-1 - 1 * -7 + 9). Let's calculate A = (5 - 6 - 7 + 6) = (5 - (6 - 7) + 6) = (5 - (-1) + 6) = (5 + 1 + 6) = 12. Let's calculate B = (-1 - 1 * -7 + 9) = (-1 - (-7) + 9) = (-1 + 7 + 9) = 15. Then, the final equation is A - B = 12 - 15 = -3. So the answer is $-3$. ✗

**LEAP** $_{\text{LOW-LEVEL}}$ **Response:** A: Let's think step by step. Let's recall that the order of operations in mathematics is as follows: (1) Parentheses, (2) exponents, (3) multiplication and division (from left to right), (4) addition and multiplication (from left to right). So, remember to always compute the expressions inside parentheses or brackets first. This equation can be written as "A - B", where A = (5 - 6 - 7 + 6) and B = (-1 - 1 * -7 + 9). Let's calculate A = (5 - 6 - 7 + 6) = (5 - 6 - 7 + 6) = (-1 - 7 + 6) = (-8 + 6) = -2. Let's calculate B = (-1 - 1 * -7 + 9) = (-1 - (-7) + 9) = (-1 + 7 + 9) = (6 + 9) = 15. Then, the final equation is A - B = -2 - 15 = -17. So the answer is $-17$. ✓

**Disambiguation QA**

**Question:** In the following sentences, explain the antecedent of the pronoun (which thing the pronoun refers to), or state that it is ambiguous. Sentence: The undergraduate applied to work with the scientist because he needed a lab assistant. Options: (A) The undergraduate needed an assistant (B) The scientist needed an assistant (C) Ambiguous

**Principles:** Pay attention to the causality and context of the sentence to identify the most likely referent for the pronoun. It is not enough to simply assume that the pronoun could refer to any possible noun mentioned in the sentence.

**CoT Response:** Let's think step by step. Here we need to determine who the pronoun "he" might be referring to. There are two possible referents for "he", namely the undergraduate and the scientist. The verb "applied" might be able to help us determine which one is more likely (if either). Let X be the undergraduate and Y the scientist. The sentence is then of the form "X applied to work with Y because (X or Y) needed a lab assistant." Let's consider Y first: "X applied to work with Y because Y needed a lab assistant." This case makes sense, because X applied to work with Y because Y had a need for a lab assistant. Now, consider X: "X applied to work with Y because X needed a lab assistant." This case also makes sense, because X applied to work with Y because X had a need for a lab assistant. Because both X and Y are possible at the same time, we conclude that the antecedent of the pronoun is ambiguous. So the answer is $(C)$. **LEAP** $_{\text{LOW-LEVEL}}$ **Response:** Let's think step by step. Here we need to determine who the pronoun "he" might be referring to. There are two possible referents for "he", namely the undergraduate and the scientist. The verb "applied" might be able to help us determine which one is more likely (if either). Let X be the undergraduate and Y the scientist. The sentence is then of the form "X applied to work with Y because (X or Y) needed a lab assistant." Let's consider Y first: "X applied to work with Y because Y needed a lab assistant." This case makes sense, because X wanted to work with Y because Y needed assistance in the lab. Now, consider X: "X applied to work with Y because X needed a lab assistant." This case doesn't make much sense, because X, as an undergraduate, would typically be the one seeking an opportunity to work with a scientist, not the other way around. Because in this context, X is the undergraduate and Y is the scientist, the answer should be the scientist. So the answer is $(B)$.

# E. Learned Principles

Tables 9-39 show the high-level principles learned by LEAP. Tables 40-43 show low-level principles learned by LEAP.

---

High-level principles (LEAP $_{\text{HIGH-LEVEL}}$ ) learned by LEAP for *boolean expressions* (BBH ) using GPT-4-0613

1. Ensure clarity and conciseness: Responses should be clear and concise to avoid confusion and maintain the user's interest.
2. Avoid redundancies: Redundant information should be eliminated to maintain the user's attention and improve the efficiency of communication.
3. Maintain uniqueness: Each response should be unique to keep the user engaged and provide a personalized experience.
4. Incorporate feedback: User feedback should be analyzed and incorporated into future responses to improve the quality of interaction.
5. Focus on relevance: Responses should be directly related to the user's query or statement to maintain relevance and provide accurate information.
6. Prioritize logical reasoning: Logical reasoning should be the foundation of all responses to ensure accuracy and credibility.
7. Preserve specific details: Specific details provided by the user should be preserved in the response to show understanding and maintain context.
8. Continual improvement: Regular introspection and analysis should be conducted to continually improve the quality of responses.

---

Table 9

---

High-level principles (LEAP $_{\text{HIGH-LEVEL}}$ ) learned by LEAP for *causal judgement* (BBH ) using GPT-4-0613

1. Consider all relevant details in the scenario: Ensure that all key details are taken into account when forming a conclusion. Do not overemphasize certain details at the expense of others.
2. Understand the concept of intentionality: It's not enough to initiate an action; the action must be carried out as planned for it to be considered intentional.
3. Consider all factors in a scenario: This includes the actor's control over the situation and the predictability of the outcome when determining intentionality.
4. Distinguish between the intent to perform an action and the intent to cause a specific outcome: In questions of causation, it's important to understand these nuances to provide accurate analysis.
5. Understand the context of causation: Intentionality involves not just the initial action but also the control and predictability of the outcome.
6. Avoid overemphasis on initial decisions: The initial decision to perform an action should not be overemphasized at the expense of other factors, such as the accidental nature of the outcome.
7. Improve understanding and interpretation of intent: This includes understanding the difference between the intent to perform an action and the intent to cause a specific outcome.
8. Ensure adequate consideration of crucial factors: In any scenario, ensure that all aspects, especially those that are crucial to the outcome, are adequately considered before arriving at a conclusion.

---

Table 10

| High-level principles (LEAP ᴴᴵᴳᴴ⁻ᴸᴱⱽᴱᴸ ) learned by LEAP for *date understanding* (BBH ) using GPT-4-0613 |
|---|
| 1. Ensure clarity and precision: Responses should be clear and concise, avoiding any ambiguity or unnecessary complexity. |

1. Ensure clarity and precision: Responses should be clear and concise, avoiding any ambiguity or unnecessary complexity.
2. Maintain relevance: The responses should be directly related to the query or topic at hand, avoiding any irrelevant information.
3. Avoid redundancy: Each response should provide unique information, avoiding repetition of previously stated facts or ideas.
4. Prioritize understanding: The responses should be designed to enhance the user's understanding of the topic, rather than simply providing information.
5. Foster engagement: The responses should be engaging and interactive, encouraging further dialogue and exploration of the topic.
6. Promote logical reasoning: The responses should be logically sound and well-reasoned, demonstrating a clear thought process.
7. Respect user's perspective: The responses should respect the user's perspective and knowledge level, avoiding any condescension or oversimplification.
8. Incorporate feedback: The responses should incorporate feedback from previous interactions, continuously improving in quality and relevance.

Table 11

| High-level principles (LEAP ᴴᴵᴳᴴ⁻ᴸᴱⱽᴱᴸ ) learned by LEAP for *disambiguation qa* (BBH ) using GPT-4-0613 |
|---|

1. The system should consider all possible interpretations of a sentence when determining the antecedent of a pronoun.
2. In ambiguous cases, the system should recognize the ambiguity and avoid making assumptions.
3. The system should consider the context and logical flow of information in the sentence to accurately determine the antecedent of a pronoun.
4. The system should improve its understanding of how information is typically exchanged in conversations or narratives.
5. The system should understand the dynamics of a conversation and the flow of information to interpret pronouns correctly.
6. The system should consider the likelihood of each potential antecedent based on the structure and semantics of the sentence.
7. The system should pay closer attention to the context and the logical sequence of events in the sentence.
8. The system should consider the roles and actions of the subjects in the sentence to accurately determine the antecedent of the pronoun.

Table 12

| High-level principles (LEAP $_{\text{HIGH-LEVEL}}$ ) learned by LEAP for *dyck languages* (BBH ) using GPT-4-0613 |
| --- |
| 1. Improve understanding of the rule that parentheses must be closed in the reverse order they were opened, which is fundamental in programming and mathematics. 
 2. Enhance the method of analyzing sequences of parentheses by processing each input one by one and keeping track of the stack configuration. 
 3. Train the model to recognize that different types of parentheses (e.g., square brackets, curly brackets, round brackets) must be matched with their corresponding closing parentheses. 
 4. Improve the model's ability to track the opening and closing of parentheses in a sequence and identify the ones that are still open at the end of the sequence. 
 5. Develop a more detailed understanding of the rules of parentheses and the ability to apply these rules to a given sequence. 
 6. Train the model to understand and apply the Last In, First Out (LIFO) principle when dealing with problems related to sequences, particularly those involving brackets or parentheses. 
 7. Ensure the model understands that the sequence ends when all brackets are properly closed, not when all opening brackets have been matched with a closing bracket. 
 8. Enhance the model's ability to correctly balance sequences of parentheses. |

Table 13

| High-level principles (LEAP $_{\text{HIGH-LEVEL}}$ ) learned by LEAP for *formal fallacies* (BBH ) using GPT-4-0613 |
| --- |
| 1. Ensure clarity and precision: Responses should be clear and concise, avoiding any ambiguity or unnecessary complexity. 
 2. Maintain relevance: The responses should be directly related to the query or topic at hand, avoiding any irrelevant information. 
 3. Avoid redundancy: Each response should provide unique information, avoiding repetition of previously stated facts or ideas. 
 4. Prioritize understanding: The responses should be designed to enhance the user's understanding of the topic, rather than simply providing information. 
 5. Foster engagement: The responses should be engaging and interactive, encouraging further discussion or exploration of the topic. 
 6. Uphold accuracy: The information provided in the responses should be accurate and reliable, based on verified sources or logical reasoning. 
 7. Adapt to context: The responses should be tailored to the specific context of the conversation, taking into account the user's knowledge level, interests, and potential biases. 
 8. Promote introspection: The responses should encourage the user to think critically and reflect on the information provided, fostering a deeper understanding and appreciation of the topic. |

Table 14

| High-level principles (LEAP HIGH-LEVEL ) learned by LEAP for *geometric shapes* (BBH ) using GPT-4-0613 |
| --- |
| 1. Improve the system's understanding of SVG path data and how it represents shapes. 
 2. Recognize that the number of "L" commands in an SVG path corresponds to the number of sides in the shape. 
 3. Understand that the "M" command in SVG path data moves the current point to a specified location without creating a line. 
 4. Count the number of "L" commands to determine the number of sides in the shape. 
 5. Recognize that the final "L" command that connects back to the initial "M" command completes the shape, and should not be counted as creating an additional side. 
 6. Improve the system's ability to interpret when an SVG path starts and ends at the same point. 
 7. Ensure that points that are visited more than once in the path, such as the starting point, are not double-counted when determining the number of sides in the polygon. 
 8. Develop a clear understanding of geometric shapes and their properties to accurately analyze and interpret SVG paths. |

Table 15

| High-level principles (LEAP HIGH-LEVEL ) learned by LEAP for *hyperbaton* (BBH ) using GPT-4-0613 |
| --- |
| 1. The system should have a comprehensive understanding of English grammar rules, including the correct order of adjectives. 
 2. The system should be able to correctly apply the identified grammatical rules to sentence analysis. 
 3. The system should be programmed to handle cases where not all categories of adjectives are present, recognizing that the order of the remaining adjectives is still important. 
 4. The system should be able to cross-verify its understanding of rules with the given options to ensure accuracy. 
 5. The system should be able to correctly identify the category each adjective falls into and ensure they are in the correct sequence. 
 6. The system should be able to correctly compare and evaluate multiple options against these rules to identify the correct answer. 
 7. The system should be improved to recognize and apply the correct adjective order of Opinion, Size, Origin, Purpose. 
 8. The system should be able to recognize when an adjective category (like Age or Color) is missing, and still maintain the correct order for the remaining adjectives. |

Table 16

| High-level principles (LEAP $_{\text{HIGH-LEVEL}}$) learned by LEAP for *logical deduction five objects* (BBH) using GPT-4-0613 |
| --- |
| 1. Enhance Interpretation Skills: The system should improve its ability to interpret and integrate multiple pieces of information accurately. 

 2. Improve Spatial Understanding: The system should be able to visualize or map out the spatial arrangement of objects based on given descriptions. 

 3. Understand Spatial Prepositions: The system should have a clear understanding of spatial prepositions and their implications in a given context. 

 4. Enhance Inference Capabilities: The system should be able to infer information that is not explicitly stated but can be logically deduced from the given information. 

 5. Eliminate Redundancies: The system should focus on capturing the essence of the feedback while eliminating any redundancies. 

 6. Maintain Clarity and Conciseness: Each point made by the system should be clear, concise, and directly derived from the introspection results. 

 7. Retain Specific Details: The system should retain specific details in its responses to ensure accuracy and completeness. 

 8. Limit Principles: The system should limit its principles to a maximum of eight to maintain focus and effectiveness. |

Table 17

| High-level principles (LEAP $_{\text{HIGH-LEVEL}}$) learned by LEAP for *logical deduction seven objects* (BBH) using GPT-4-0613 |
| --- |
| 1. Ensure clarity and precision: Responses should be clear and concise, avoiding any ambiguity or unnecessary complexity. 

 2. Maintain relevance: Responses should directly address the query or topic at hand, avoiding any unrelated or tangential information. 

 3. Prioritize uniqueness: Strive to provide unique insights or perspectives in responses, avoiding repetition or common knowledge. 

 4. Emphasize logical reasoning: Responses should be logically sound and well-reasoned, with each point building on the last to form a coherent argument or explanation. 

 5. Eliminate redundancies: Avoid repeating the same information or points in a response, as this can dilute the message and make the response less engaging. 

 6. Incorporate specific details: Where relevant, include specific details in responses to add depth and richness to the information provided. 

 7. Limit response length: Keep responses to a reasonable length to ensure they are digestible and maintain the reader's interest. 

 8. Continually improve: Regularly review and analyze responses to identify areas for improvement and implement changes as necessary. |

Table 18

High-level principles (LEAP $_{\text{HIGH-LEVEL}}$ ) learned by LEAP for *logical deduction three objects* (BBH ) using GPT-4-0613

1. Ensure clarity and precision: Responses should be clear and concise, avoiding any ambiguity or unnecessary complexity.
2. Maintain relevance: Responses should directly address the query or topic at hand, avoiding any unrelated or tangential information.
3. Prioritize uniqueness: Strive to provide unique insights or perspectives in responses, avoiding repetition or common knowledge.
4. Foster engagement: Responses should be engaging and interesting, aiming to stimulate further discussion or thought.
5. Uphold accuracy: Ensure all information provided is accurate and up-to-date, avoiding any misinformation or outdated facts.
6. Promote comprehensibility: Use language and terminology that is easily understood by the intended audience, avoiding jargon or overly complex language.
7. Encourage brevity: Keep responses succinct and to the point, avoiding unnecessary length or verbosity.
8. Respect context: Consider the context of the query or discussion when formulating responses, ensuring they are appropriate and relevant.

Table 19

High-level principles (LEAP $_{\text{HIGH-LEVEL}}$ ) learned by LEAP for *movie recommendation* (BBH ) using GPT-4-0613

1. Ensure Consistency in Labeling: Always cross-check the labels used in the reasoning process with those in the question to avoid discrepancies that could lead to incorrect conclusions.
2. Comprehensive Analysis: Consider all aspects of the subject matter, such as time period, cultural significance, and popularity, not just the most obvious or immediate ones.
3. Contextual Understanding: Always consider the broader context of the question to avoid focusing too narrowly on one aspect. This will help in generating more accurate and relevant answers.
4. Avoid Redundancies: Strive to eliminate any redundancies in the reasoning process to maintain clarity and precision.
5. Emphasize Uniqueness: Ensure that each principle is unique and offers a different perspective or approach to improve the reasoning process.
6. Clarity and Conciseness: Make sure each principle is clear, concise, and directly derived from the introspection results.
7. Detail Orientation: Do not omit specific details that could be crucial to the reasoning process.
8. Continuous Improvement: Regularly review and update the principles based on new insights or feedback to ensure continuous improvement in the reasoning process.

Table 20

---

High-level principles (LEAP HIGH-LEVEL ) learned by LEAP for *multistep arithmetic two* (BBH ) using GPT-4-0613

---

1. Strictly adhere to the order of operations (PEMDAS/BODMAS) when performing calculations.
2. Correctly apply the rules of arithmetic, including multiplication and subtraction operations.
3. Handle negative numbers accurately, understanding that the multiplication of two negative numbers results in a positive number.
4. Interpret mathematical notation correctly, especially when dealing with negative numbers and subtraction operations.
5. Understand that subtraction of a negative number is equivalent to addition of the absolute value of that number.
6. Double-check intermediate results to ensure accuracy and avoid simple arithmetic errors.
7. Ensure that the system is designed to correctly perform operations within parentheses.
8. Improve handling of unique cases, such as double negatives, to avoid misinterpretation.

---

Table 21

---

High-level principles (LEAP HIGH-LEVEL ) learned by LEAP for *navigate* (BBH ) using GPT-4-0613

---

1. Ensure clarity and precision: Responses should be clear, concise, and directly answer the question or statement at hand. Avoid ambiguity or overly complex language.
2. Maintain relevance: Stay on topic and ensure that the response is directly related to the query or statement. Avoid going off on tangents or introducing unrelated information.
3. Avoid redundancy: Each response should provide new information or a unique perspective. Avoid repeating the same points or ideas.
4. Be insightful: Responses should provide meaningful insights or perspectives. They should not merely restate the obvious or provide generic answers.
5. Use specific details: When appropriate, include specific details in responses to enhance understanding and provide context. Avoid being too vague or general.
6. Be logical: Responses should follow a logical structure and reasoning. Avoid inconsistencies or contradictions in the response.
7. Adapt to the context: The tone, language, and content of the response should be appropriate for the context. Avoid using inappropriate or irrelevant language or content.
8. Be respectful: Always maintain a respectful and professional tone in responses. Avoid offensive or disrespectful language or content.

---

Table 22

| High-level principles (LEAP $_{\text{HIGH-LEVEL}}$ ) learned by LEAP for *object counting* (BBH ) using GPT-4-0613 |
| --- |
| 1. Ensure clarity and precision: Responses should be clear, concise, and directly derived from the information provided. Avoid ambiguity and ensure the response is easily understood by the user. 
 2. Eliminate redundancies: Avoid repeating the same information in different ways. Each response should provide new, unique insights. 
 3. Focus on relevance: The response should be directly related to the user's query or statement. Irrelevant information can confuse the user and detract from the overall quality of the response. 
 4. Maintain logical consistency: The response should follow a logical flow based on the user's input. Avoid contradicting previous statements or information. 
 5. Prioritize user's needs: The response should be tailored to the user's needs and preferences. Consider the context and purpose of the user's query when formulating the response. 
 6. Incorporate feedback: Regularly analyze user feedback to identify areas for improvement. Use this feedback to refine and enhance future responses. 
 7. Promote engagement: The response should encourage further interaction from the user. This can be achieved by asking follow-up questions or providing additional relevant information. 
 8. Uphold ethical standards: Ensure that the response respects the user's privacy and adheres to ethical guidelines. Avoid providing information that could potentially harm the user or others. |

Table 23

| High-level principles (LEAP $_{\text{HIGH-LEVEL}}$ ) learned by LEAP for *penguins in a table* (BBH ) using GPT-4-0613 |
| --- |
| 1. Comprehensive Data Analysis: The system should thoroughly analyze all data points to ensure no relevant information is missed. 
 2. Accurate Identification: The system should correctly identify all instances that meet the given criteria. 
 3. Counting Accuracy: The system should accurately count all instances that meet the given criteria. 
 4. Data Interpretation: The system should improve its ability to interpret data correctly to ensure accurate results. 
 5. Error Correction: The system should have mechanisms in place to correct errors in data analysis and interpretation. 
 6. Continuous Improvement: The system should continuously strive to improve its performance based on feedback and introspection results. 
 7. Redundancy Elimination: The system should focus on capturing the essence of the data, eliminating redundancies where possible. 
 8. Clarity and Conciseness: The system should ensure that its responses are clear, concise, and directly derived from the data. |

Table 24

High-level principles (LEAP $_{\text{HIGH-LEVEL}}$ ) learned by LEAP for *reasoning about colored objects* (BBH ) using GPT-4-0613

1. Ensure clarity and precision: Responses should be clear and concise, avoiding any ambiguity or unnecessary complexity.
2. Maintain relevance: The responses should be directly related to the query or topic at hand, avoiding any irrelevant information.
3. Avoid redundancy: Each response should provide unique information, avoiding repetition of previously stated facts or ideas.
4. Prioritize understanding: The responses should be designed to enhance the user's understanding of the topic, rather than simply providing information.
5. Foster engagement: The responses should be engaging and interactive, encouraging further dialogue and exploration of the topic.
6. Uphold accuracy: The information provided in the responses should be accurate and reliable, based on verified sources.
7. Adapt to user's needs: The responses should be tailored to the user's level of knowledge and interest, providing more detailed information when necessary and simplifying complex concepts when appropriate.
8. Promote introspection: The responses should encourage the user to think critically about the topic, promoting deeper understanding and personal growth.

Table 25

High-level principles (LEAP $_{\text{HIGH-LEVEL}}$ ) learned by LEAP for *ruin names* (BBH ) using GPT-4-0613

1. Ensure clarity and conciseness: Responses should be clear and concise to avoid any confusion or misunderstanding. Avoid using complex language or jargon that may not be understood by all users.
2. Eliminate redundancies: Avoid repeating the same information in different ways. This can make responses longer than necessary and can be confusing for the user.
3. Directly derive from introspection results: Responses should be directly based on the analysis or introspection results. This ensures that the responses are relevant and accurate.
4. Capture the essence of the feedback: The main points or key messages of the feedback should be captured in the responses. This ensures that the responses are meaningful and valuable to the user.
5. Limit the number of principles: Too many principles can be overwhelming and difficult to remember. Limit the number of principles to a maximum of 8.
6. Leave specific details in place: While it's important to be concise, it's also important not to remove specific details that are necessary for understanding the response. These details can provide context and depth to the response.
7. Ensure uniqueness: Each principle should be unique and not overlap with other principles. This ensures that each principle provides a distinct value.
8. Focus on improvement: The principles should focus on ways to improve future responses. This ensures that the principles are forward-looking and proactive.

Table 26

High-level principles (LEAP $_{\text{HIGH-LEVEL}}$ ) learned by LEAP for *salient translation error detection* (BBH ) using GPT-4-0613

1. The system should enhance its focus on the overall meaning and context of the sentence, not just the presence or absence of certain words or phrases.
2. The system should improve its ability to identify and classify errors accurately, particularly those related to dropped content.
3. The system should pay more attention to Named Entities errors, ensuring that the entity in the translation matches the entity in the source.
4. The system should improve its precision in identifying the number of errors present in a translation, avoiding overestimation or underestimation.
5. The system should enhance its understanding of the context and specific details within the text to better identify and differentiate between different types of errors.
6. The system should improve its ability to accurately translate and compare the original and translated texts to identify any discrepancies.
7. The system should be able to recognize when a specific name, place, or location is altered in the translation, which constitutes a named entity error.
8. The system should focus on improving its ability to identify errors related to named entities, rather than misclassifying them as other types of errors.

Table 27

High-level principles (LEAP $_{\text{HIGH-LEVEL}}$ ) learned by LEAP for *snarks* (BBH ) using GPT-4-0613

1. Ensure clarity and precision: Responses should be clear and concise, avoiding any ambiguity or confusion.
2. Eliminate redundancies: Avoid repeating the same information or ideas in different ways.
3. Direct derivation: Each response should be directly derived from the information provided, without making unnecessary assumptions or inferences.
4. Maintain uniqueness: Each response should provide unique insights, rather than reiterating common knowledge or previously stated information.
5. Focus on relevance: Responses should be directly relevant to the question or topic at hand, avoiding any irrelevant or off-topic information.
6. Prioritize insightful content: Responses should aim to provide new insights or perspectives, rather than simply restating the obvious.
7. Respect the context: The context in which the question or topic is presented should be taken into account when formulating responses.
8. Preserve specific details: While responses should be concise, they should not omit important details that are necessary for a full understanding of the topic.

Table 28

High-level principles (LEAP $_{\text{HIGH-LEVEL}}$ ) learned by LEAP for *sports understanding* (BBH ) using GPT-4-0613

1. Understand the Context: The system should focus on understanding the context and the exact question being asked, differentiating between different types of questions and providing reasoning accordingly.
2. Accurate Knowledge: The system should have accurate and up-to-date knowledge about the subject matter. In this case, understanding the rules of basketball and the correct terminology is crucial.
3. Avoid Unstated Assumptions: The system should be careful not to make assumptions that are not explicitly stated in the sentence or question.
4. Consider Structure: The system should consider the structure of the subject matter for accurate analysis. For example, understanding the structure of the NBA Finals, which includes teams from both conferences.
5. Adapt Response Complexity: The system should adapt its responses based on the complexity and detail level of the question or task at hand. A simpler, more direct response may be more appropriate in some cases.
6. Consider All Plausible Scenarios: The system should consider all plausible scenarios, not just the most common or current ones. For example, the possibility of player trades in the NBA.
7. Update Information: The system should be updated with the most recent and accurate information about the subject matter to provide the most accurate analysis.
8. Unique and Insightful Principles: The system should generate unique and insightful principles to improve future responses, focusing on capturing the essence of the feedback while eliminating redundancies.

Table 29

High-level principles (LEAP $_{\text{HIGH-LEVEL}}$ ) learned by LEAP for *temporal sequences* (BBH ) using GPT-4-0613

1. Ensure clarity and precision: Responses should be clear and concise, avoiding any ambiguity or confusion.
2. Eliminate redundancies: Avoid repeating the same information or ideas in different ways.
3. Direct derivation: Each response should be directly derived from the information provided, without making unnecessary assumptions or inferences.
4. Maintain uniqueness: Each response should provide unique insights, rather than reiterating common knowledge or previously stated information.
5. Focus on relevance: Responses should be directly relevant to the question or topic at hand, avoiding any irrelevant or off-topic information.
6. Prioritize insightful content: Responses should aim to provide new insights or perspectives, rather than simply restating the obvious.
7. Respect the context: The context in which the question or topic is presented should be taken into account when formulating responses.
8. Keep the audience in mind: The responses should be tailored to the needs and expectations of the intended audience.

Table 30

High-level principles (LEAP HIGH-LEVEL) learned by LEAP for *tracking shuffled objects five objects* (BBH) using GPT-4-0613

1. Ensure clarity and precision: Responses should be clear and concise, avoiding any ambiguity or unnecessary complexity.
2. Maintain relevance: The responses should be directly related to the query or topic at hand, avoiding any irrelevant information or digressions.
3. Prioritize uniqueness: Each response should offer a unique perspective or insight, avoiding repetition or redundancy.
4. Incorporate feedback: Future responses should take into account any feedback received, using it to improve the quality and relevance of the responses.
5. Focus on logic: Responses should be logically sound, with each point or argument following logically from the previous one.
6. Be insightful: Responses should aim to provide new insights or perspectives, rather than simply reiterating known information.
7. Maintain consistency: The tone, style, and content of the responses should be consistent, ensuring a coherent and cohesive narrative.
8. Respect context: The responses should take into account the context in which the query or topic is being discussed, ensuring that they are appropriate and relevant.

Table 31

High-level principles (LEAP HIGH-LEVEL) learned by LEAP for *tracking shuffled objects seven objects* (BBH) using GPT-4-0613

1. Ensure clarity and precision: Responses should be clear and concise, avoiding any ambiguity or unnecessary complexity.
2. Maintain relevance: The responses should be directly related to the query or topic at hand, avoiding any irrelevant information or digressions.
3. Prioritize uniqueness: Strive to provide unique insights or perspectives in the responses, avoiding repetition or common knowledge.
4. Foster engagement: Responses should be engaging and interesting, aiming to stimulate further discussion or thought.
5. Uphold accuracy: Ensure that all information provided in the responses is accurate and up-to-date, avoiding any misinformation or outdated facts.
6. Promote comprehensibility: Use simple and understandable language in the responses, ensuring they are accessible to a wide range of audiences.
7. Encourage brevity: Keep responses succinct and to the point, avoiding unnecessary length or verbosity.
8. Emphasize logical reasoning: Responses should be logically sound and well-reasoned, ensuring they are credible and persuasive.

Table 32

High-level principles (LEAP $_{\text{HIGH-LEVEL}}$ ) learned by LEAP for *tracking shuffled objects three objects* (BBH ) using GPT-4-0613

1. Accurate Tracking: The system should accurately track changes in a sequence of events to ensure correct interpretation and response.
2. Avoid Assumptions: The system should not make assumptions that are not supported by the given information. It should only rely on the facts presented.
3. Strict Adherence: The system should strictly adhere to the described events and changes, without deviating from the provided information.
4. Partner Swaps: In the context of a dance, the system should correctly follow partner swaps to ensure no dancer is left without a partner.
5. Data Verification: The system should verify the data it receives to ensure it is accurate and reliable before making any decisions or predictions.
6. Continuous Improvement: The system should continuously learn and improve its performance based on feedback and analysis of past responses.
7. Contextual Understanding: The system should understand the context of the information provided to make accurate interpretations.
8. Error Correction: The system should have mechanisms in place to correct errors and inaccuracies in its responses.

Table 33

High-level principles (LEAP $_{\text{HIGH-LEVEL}}$ ) learned by LEAP for *web of lies* (BBH ) using GPT-4-0613

1. The AI model should always pay close attention to the format of the question.
2. The model should provide the answer in the same format as the question to ensure consistency.
3. If the question is a yes/no question, the answer should be given as "Yes" or "No" to directly address the question.
4. The model should be trained to understand the format of the question and provide the answer in the same format.
5. The model's responses should align with the expectations set by the question.
6. The answer should be directly relevant to the question and easy for the user to understand.
7. The model should match the style of the answer to the style of the question.
8. In a conversational style question, the answer should also be given in a conversational style.

Table 34

High-level principles (LEAP $_{\text{HIGH-LEVEL}}$ ) learned by LEAP for *word sorting* (BBH ) using GPT-4-0613

1. Maintain the application of the principle of alphabetical sorting in future tasks.
2. Prepare the model to handle more complex cases, such as words with identical prefixes.
3. Ensure the model can handle words that differ in case or punctuation.
4. Continually assess the model's understanding and application of alphabetical order.
5. Maintain the model's level of understanding and accuracy for similar tasks.
6. Expose the model to a variety of tasks to ensure it can generalize the principle across different contexts.
7. Ensure the model can handle sorting words with special characters or numbers.
8. Regularly evaluate and improve the model's performance based on feedback and introspection results.

Table 35

High-level principles (LEAP $_{\text{HIGH-LEVEL}}$ ) learned by LEAP for DROP using Gemini Pro

1. Maintain precision when dealing with percentages to avoid rounding errors.
2. Perform calculations using the full precision available and only round as a final step, if necessary.
3. Avoid rounding when answering questions that ask for exact values.
4. Consider the level of rounding carefully to ensure that it does not introduce significant errors.
5. Rounding can lead to incorrect conclusions and misinterpretations of data.
6. Keep the original values and perform calculations using the full precision available.
7. Rounding should be done only after all calculations are complete.
8. Rounding errors can lead to incorrect results.

Table 36

High-level principles (LEAP $_{\text{HIGH-LEVEL}}$ ) learned by LEAP for DROP using GPT-4-0613

1. Improve context understanding: The system should be designed to better understand the context of the question, including specific details and requirements, to generate accurate answers.
2. Prioritize direct information: The system should prioritize direct information given in the passage over inferred information to avoid unnecessary assumptions or calculations.
3. Accurate interpretation: The system should carefully interpret the context of the question, recognizing specific details and changes in entities or events over time.
4. Adhere to the required format: The system should ensure that the final answer adheres to the required format, such as providing a single numerical answer when required.
5. Avoid unnecessary assumptions: The system should not make unnecessary assumptions when the required information is directly provided in the passage.
6. Improve accuracy of analysis: The system should strive to improve the accuracy of its analysis by focusing on the specific details provided in the passage and the question.
7. Recognize transitions and transformations: The system should be capable of recognizing when one entity transitions or transforms into another, as this can affect the interpretation of the question and the accuracy of the answer.
8. Focus on direct answers: When the answer is directly stated in the passage, the system should focus on providing that answer rather than attempting to infer or calculate the answer.

Table 37

High-level principles (LEAP $_{\text{HIGH-LEVEL}}$ ) learned by LEAP for HotpotQA using Gemini Pro

1. Verify the accuracy of information before drawing conclusions, especially when comparing start dates of entities.
2. Consider all relevant aspects when identifying commonalities between entities, not just the most obvious or prominent ones.
3. Analyze the work of mathematicians or scientists within the broader context of their field of study, rather than focusing solely on specific subfields or topics.
4. Consider all common professions when answering questions about the professions of multiple people, not just the most prominent one.
5. Avoid making erroneous conclusions based on incorrect data.
6. Ensure a comprehensive analysis of all shared characteristics to provide accurate and complete answers about commonalities.
7. Provide a more accurate assessment of similarities and differences in the work of mathematicians or scientists by considering the broader field of study.
8. Eliminate redundancies and capture the essence of the feedback to create clear, concise, and directly derived principles.

Table 38

High-level principles (LEAP $_{\text{HIGH-LEVEL}}$ ) learned by LEAP for HotpotQA using GPT-4-0613

1. Ensure comprehensive analysis of all relevant information, considering all aspects of the subject matter, not just the most prominent ones.
2. Improve categorization of work or field of study, understanding the hierarchical and categorical relationships between different fields.
3. Always conclude responses with a clear answer statement, such as "So the answer is <answer>".
4. Expand the range of accessible information to answer specific questions accurately.
5. Improve understanding of the topic at hand, pulling from a wider range of data if necessary.
6. Recognize when a question is asking for a specific piece of information and strive to provide that.
7. Avoid defaulting to "unknown" when the answer isn't immediately apparent, instead, make an effort to find the required information.
8. Improve the ability to recognize and understand the background and inspirations of characters or subjects in various fields.

Table 39

Low-level principles (LEAP $_{\text{LOW-LEVEL}}$ ) learned by LEAP for HotpotQA using GPT-4-0613

1. The principle that should be looked at carefully to improve the performance in the future is to ensure a comprehensive analysis of all the relevant information. In this case, all the professions of the individuals should have been considered, not just the most prominent one. This will help to provide a more accurate and complete answer.
2. The system should be more careful when categorizing the type of work or field of study of individuals. Even if their specific areas of focus or contributions differ, they may still belong to the same broader field or type of work. In this case, both individuals are mathematicians, so they are known for the same type of work. The system should be able to recognize and understand the hierarchical and categorical relationships between different fields of study or types of work.
3. Pay attention to the format! End your response with "So the answer is <answer>".
4. The system should be able to access a broader range of information to answer specific questions accurately. In this case, it would need to know about the background and inspirations of Matt Groening's characters in 'The Simpsons'. This suggests that the system needs to be able to pull from a wider range of data or have a more comprehensive understanding of the topic at hand. It's also important for the system to recognize when a question is asking for a specific piece of information and to strive to provide that, rather than defaulting to "unknown" when the answer isn't immediately apparent.

Table 40

Low-level principles (LEAP $_{\text{LOW-LEVEL}}$ ) learned by LEAP for HotpotQA using Gemini Pro

1. When comparing the start dates of two entities, it is crucial to ensure the accuracy of the information used. Incorrect data can lead to erroneous conclusions. Always verify the accuracy of information before drawing conclusions.
2. When answering questions about commonalities between two entities, it is important to consider all relevant aspects and not just the most obvious or prominent one. A comprehensive analysis of all shared characteristics is necessary to provide an accurate and complete answer.
3. When analyzing the work of mathematicians or scientists, it is important to consider the broader field of study that their contributions belong to, rather than focusing solely on the specific subfields or topics that they worked on. This broader perspective allows for a more accurate assessment of the similarities and differences in their work and helps to avoid incorrect conclusions about the nature of their contributions.
4. When answering questions about the professions of multiple people, it is important to consider all of the professions that they have in common, not just the most prominent one.

Table 41

---

Low-level principles (LEAP $_{\text{LOW-LEVEL}}$) learned by LEAP for DROP using GPT-4-0613

---

1. The system should be designed to understand the context of the question better. In this case, it should have recognized that the question was asking for the duration of existence of the European Coal and Steel Community before it transitioned into the European Economic Community. Understanding the specific context and requirements of a question is crucial for generating accurate answers.

2. The system should be designed to carefully interpret the context of the question. In this case, it should have recognized that the ECSC transformed into the EEC in 1958, and therefore, its existence as the ECSC ended in that year. Understanding the context and specific details of the question is crucial for generating accurate answers.

3. The system should prioritize direct information given in the passage over inferred information. In this case, the direct information was the total number of touchdown passes thrown by Stafford, which was clearly stated in the passage. The system should not make unnecessary assumptions or calculations when the required information is directly provided. This will help to avoid errors and improve the accuracy of the analysis.

4. Pay attention to the format! Make sure your final answer should be a single numerical number, in the form boxedanswer, at the end of your response.

---

Table 42

---

Low-level principles (LEAP $_{\text{LOW-LEVEL}}$) learned by LEAP for DROP using Gemini Pro

---

1. When dealing with percentages, it is important to maintain precision and avoid rounding errors, especially when the difference between values is small. Rounding should only be done as a final step, if necessary, to ensure that the result is presented in a clear and concise manner.

2. When dealing with percentages, it is important to maintain precision and avoid rounding errors. Rounding should only be done as a final step, after all calculations are complete.

3. When dealing with percentages, it is important to maintain precision and avoid rounding errors. Rounding can lead to incorrect conclusions and misinterpretations of data. Always keep the original values and perform calculations using the full precision available. Rounding should only be done as a final step, if necessary, and the level of rounding should be carefully considered to ensure that it does not introduce significant errors.

4. When answering questions that ask for exact values, it is important to not round the answer unless specifically instructed to do so. Rounding the answer can lead to incorrect results, as seen in this example.

---

Table 43

---

| Dataset | CoT | LLL | HLL | Winner |
|---|---|---|---|---|
| boolean_expressions | **79.47** | 76.8 | 62.13 | CoT |
| causal_judgement | **64.35** | 50.8 | 56.15 | CoT |
| date_understanding | **81.2** | 0.27 | 14.0 | CoT |
| disambiguation_qa | 1.2 | 9.47 | **26.4** | LEAP HIGH-LEVEL |
| dyck_languages | 30.0 | 14.0 | **30.67** | LEAP HIGH-LEVEL |
| formal_fallacies | **54.0** | 49.6 | 44.0 | CoT |
| geometric_shapes | **2.4** | 0.0 | 0.0 | CoT |
| hyperbaton | 51.6 | 46.27 | **53.07** | LEAP HIGH-LEVEL |
| logical_deduction_five_objects | **26.67** | 1.33 | 0.0 | CoT |
| logical_deduction_seven_objects | 14.13 | **15.33** | 0.0 | LEAP LOW-LEVEL |
| logical_deduction_three_objects | **22.53** | 16.67 | 3.07 | CoT |
| movie_recommendation | **59.2** | 1.33 | 7.6 | CoT |
| multistep_arithmetic_two | 10.53 | 10.13 | **11.73** | LEAP HIGH-LEVEL |
| navigate | **76.4** | 56.4 | 62.53 | CoT |
| object_counting | 60.4 | 59.73 | **60.53** | LEAP HIGH-LEVEL |
| penguins_in_a_table | **31.05** | 26.94 | 27.4 | CoT |
| reasoning_about_colored_objects | **38.93** | 9.47 | 5.07 | CoT |
| ruin_names | **0.0** | 0.0 | 0.0 | CoT |
| salient_translation_error_detection | **30.13** | 2.67 | 22.53 | CoT |
| snarks | **60.11** | 57.68 | 30.34 | CoT |
| sports_understanding | **82.4** | 4.13 | 51.47 | CoT |
| temporal_sequences | 4.53 | **16.8** | 10.13 | LEAP LOW-LEVEL |
| tracking_shuffled_objects_five_objects | **16.4** | 6.13 | 2.8 | CoT |
| tracking_shuffled_objects_seven_objects | **15.07** | 13.2 | 9.33 | CoT |
| tracking_shuffled_objects_three_objects | **23.6** | 13.33 | 16.67 | CoT |
| web_of_lies | **99.6** | 84.53 | 96.8 | CoT |
| word_sorting | 19.33 | 11.33 | **23.87** | LEAP HIGH-LEVEL |

Table 44: BBH **Results:** Comparative performance results on various BBH datasets using Llama-2-chat-70B . With Llama-2-chat-70B , LEAP underperforms vanilla CoT . An analysis of outputs reveals that the present of lessons causes Llama-2-chat-70B to hallucinate bad output at a dramatically high rates for some tasks. For example, instead of generating the answer, Llama-2-chat-70B + LEAP will start repeating the instructions to format the answer or start generating new samples. This indicates that instruction tuning for Llama-2-chat-70B might be brittle.

## F. Open-Source Models Experiments

We also experimented with Llama-2-chat-70B to investigate how the open-access models perform with LEAP. We find mixed results in experimenting with Llama-2-chat-70B for LEAP. Specifically, we find that for BBH tasks, LEAP is ineffective in improving reasoning performance. The lessons generated by LEAP overwhelm Llama-2-chat-70B , indicating that the ability to follow and act on complex instructions might be the key.

```
Instruction: {instruction}

Question: {question}

End your answer with "So the answer is <answer>."

Think step by step.
```

Figure 6: Prompt template for the mistake generation step.

```
Question: {question}

Generated Reasoning: {response}

Generated Answer: {generated_answer}

Correct Reasoning: {correct_reasoning}

Correct Answer: {correct_answer}

Instruction: Conduct a thorough analysis of the generated answer in comparison to the
correct answer. Also observe how the generated reasoning differs from the correct
reasoning. Identify any discrepancies, misunderstandings, or errors. Provide clear
insights, principles, or guidelines that can be derived from this analysis to improve
future responses. We are not focused on this one data point, but rather on the general
principle.

Reasoning: <discuss why the generated answer is wrong>
Insights: <what principle should be looked at carefully to improve the performance in
the future>
```

Figure 7: Prompt template for the low-level principle learning step.

```
Low-level principles: {low_level_principles}

Create a list of *unique* and insightful principles to improve future responses based
on the analysis above.
Focus on capturing the essence of the feedback while eliminating redundancies.
Ensure that each point is clear, concise, and directly derived from the introspection
results.
Create a numbered list of principles. Leave specific details in place.
Limit to at most 8 principles.

List of Principles:
```

Figure 8: Prompt template for the high-level principle learning step.

## G. Prompts

This section contains the prompts for the different steps in LEAP: Figure 6 shows the prompt template for the first step of *generating mistakes*; Figure 7 shows the prompt template for the second step of generating low-level principles ($\mathcal{L}_{\text{LOW-LEVEL}}$); Figure 8 shows the prompt template for the third step of generating high-level principles ($\mathcal{L}_{\text{HIGH-LEVEL}}$); finally, Figure 9 shows the prompt template for the final inference on unseen examples.

```
Instruction: {instruction}
In doing so, please carefully note the following principles:

Principles: {principles}

---

{few_shot_questions_and_answers}

Q: {test_question}
```

Figure 9: Prompt template for the final step of inference on unseen example, using either low- or high-level principles.

