# OpenReview forum: "In-Context Principle Learning from Mistakes"
_ICML.cc/2024/Workshop/ICL — ICML 2024 Workshop ICL Poster_

### Official Review · Reviewer_c5o1 · 2024-05-29
**Review to Paper In-Context Principle Learning from Mistakes**

**Rating:** 2
**Fit:** 3
**Confidence:** 2

**Workshop Review:**

- Introduce LEAP (Learning Principles) where we induce model to reflect on mistakes and learn explicit task-specific principles from them. LEAP acts by given incorrect CoT response for a specific input-output example, it generates explicit priniciples by providing LLM with mistaken CoT with correct output.
- Strengths
    - Diving LEAP into 2 different granularity levels: high-level and low-level is interesting.
    - It was easy to follow the flow of the paper with many qualitative examples shown to support their hypothesis.
- Weaknesses
    - The idea of prompting LLM to generate reasoning for “why they got the incorrect answer” is not a very novel idea. Also, learning from
    - The results are not very effective, there still remains some variation (model or benchmark) where few-shot CoT is better than LEAP while LEAP needs additional round of reasoning (generating reasoning and answers) using costly closed-source LLMs.
    - More importantly, results with Llama-2 70b shows that the effectiveness of LEAP does not extend to open-sourced LLMs where few-shot CoT always outperform LEAP.
    - Take together, this suggests that LEAP requires a base LLM with strong reasoning capability mostly which are closed-source LLMs. However, to the reviewer, the effectiveness of LEAP is not strong enough to convince that we need an additional step of expensive reasoning.
    - When looking at the results tables in the Appendix, it also supports that in many cases, LEAP is not effective than few-shot CoT.

**Reason For Not Giving Higher Score:**

N/A

**Reason For Not Giving Lower Score:**

N/A

---

### Official Review · Reviewer_MoNo · 2024-06-03
**Review of Submission52**

**Rating:** 2
**Fit:** 3
**Confidence:** 2

**Workshop Review:**

This paper proposes a novel in-context learning method called LEAP. The method intentionally introduces mistakes into few-shot prompting and instructs the model to derive task-specific principles from these errors. These principles help the model avoid similar mistakes in future tasks. Experiments demonstrate that LEAP effectively improves the performance of LLMs across various reasoning benchmarks. The paper is clearly written and well-structured, making it accessible to readers. The method’s novelty lies in its approach to learning from incorrect examples, which is valuable to the in-context learning community.

**Reason For Not Giving Higher Score:**

The proposed method seems not applicable to tasks where there is no ground-truth answer, such as writing. Additionally, since the method shares high-level ideas with self-correcting errors, the experiments could be strengthened by comparing LEAP with methods in this line of work, such as [1-3].

[1] Self-refine: Iterative refinement with self-feedback
[2] Tree of thoughts: Deliberate problem solving with large language models
[3] Self-consistency improves chain of thought reasoning in language models

**Reason For Not Giving Lower Score:**

The proposed method is interesting and effective, addressing a relevant topic in the in-context learning community.

---

### Meta-Review · Area_Chair_KkHt · 2024-06-16

**Recommendation:** 2

**Metareview:**

The paper introduces an in-context learning method (LEAP) that prompts LLMs to reflect on their mistakes and derive explicit task-specific principles from them. LEAP operates by generating explicit principles from incorrect chain-of-thought (CoT) responses and correct outputs, divided into high-level and low-level granularity. Reviewers appreciated the clear structure and qualitative examples supporting the hypothesis. However, they noted that LEAP's effectiveness is inconsistent. Specifically, LEAP requires additional rounds of reasoning using costly closed-source LLMs and does not perform as well with open-source models like Llama-2 70b, where few-shot CoT often outperforms LEAP. Additionally, the method's applicability is limited in tasks without ground-truth answers, and comparisons with similar self-correcting methods could strengthen the experiments. Despite these limitations, the paper's innovative approach to learning from incorrect examples and its clear alignment with the workshop's focus makes it a valuable contribution, suggesting acceptance.

---

### Decision · Program_Chairs · 2024-06-17

Accept (Poster)